# LabelAny3D: Label Any Object 3D in the Wild

Jin Yao[*]   Radowan Mahmud Redoy[*]   Sebastian Elbaum   Matthew B. Dwyer   Zezhou Cheng

University of Virginia

https://uva-computer-vision-lab.github.io/LabelAny3D/

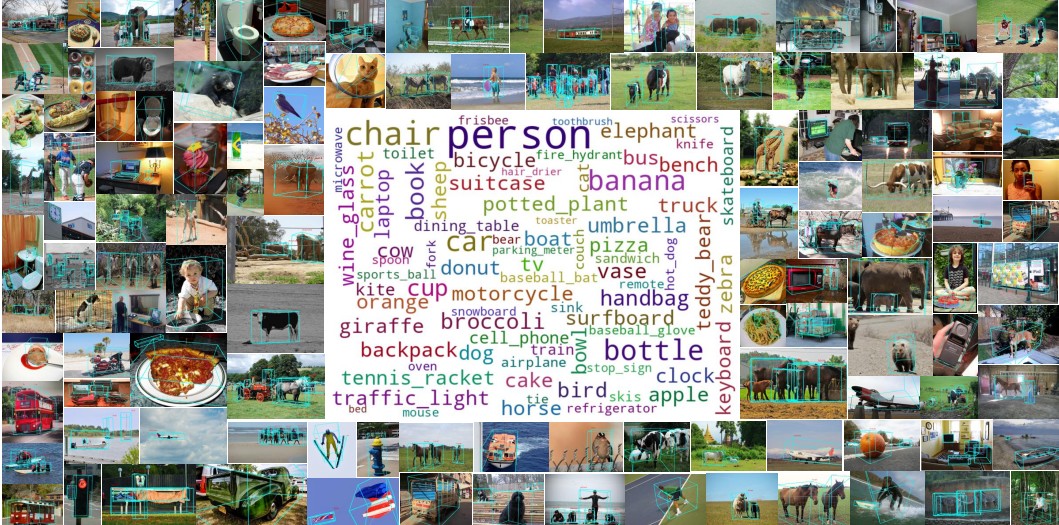

Figure 1: Samples from our proposed COCO3D benchmark.

## Abstract

Detecting objects in 3D space from monocular input is crucial for applications ranging from robotics to scene understanding. Despite advanced performance in the indoor and autonomous driving domains, existing monocular 3D detection models struggle with in-the-wild images due to the lack of 3D in-the-wild datasets and the challenges of 3D annotation. We introduce **LabelAny3D**, an *analysis-by-synthesis* framework that reconstructs holistic 3D scenes from 2D images to efficiently produce high-quality 3D bounding box annotations. Built on this pipeline, we present **COCO3D**, a new benchmark for open-vocabulary monocular 3D detection, derived from the MS-COCO dataset and covering a wide range of object categories absent from existing 3D datasets. Experiments show that annotations generated by LabelAny3D improve monocular 3D detection performance across multiple benchmarks, outperforming prior auto-labeling approaches in quality. These results demonstrate the promise of foundation-model-driven annotation for scaling up 3D recognition in realistic, open-world settings.

## 1   Introduction

Monocular 3D object detection—recognizing and localizing objects in 3D space from a single RGB image—is an emerging research direction with wide-ranging applications in robotics, autonomous driving, and embodied AI. Compared to approaches that rely on specialized sensors such as LiDAR

---

[*]Equal contribution

39th Conference on Neural Information Processing Systems (NeurIPS 2025).

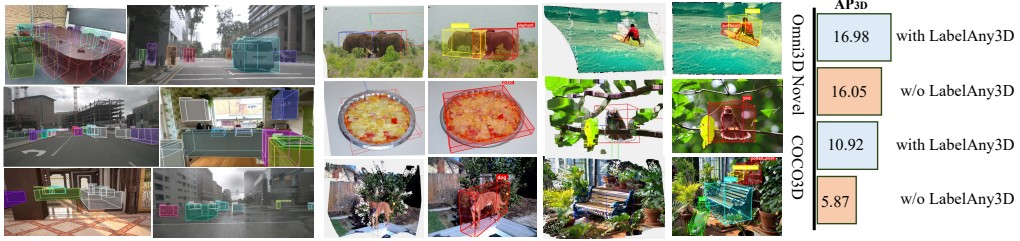

(a) Omni3D: indoor or self-driving     (b) 3D annotations from LabelAny3D     (c) Monocular 3D Detection

Figure 2: **Overview. (a)** Omni3D [7] offers large-scale 3D annotations but primarily covers indoor and self-driving scenarios. **(b)** The proposed LabelAny3D reconstructs 3D scenes (left) to annotate objects in 3D (right). **(c)** Leveraging LabelAny3D pseudo-labels to train a monocular 3D detector raises its $AP_{3D}$ (average precision; higher is better) on both Omni3D novel categories [75] and our new COCO3D benchmark.

or multi-view stereo [46, 47, 11, 12, 77, 84, 80, 79, 52, 20], monocular methods are lightweight, accessible, and energy-efficient, making them well-suited for many real-world deployments (*e.g.*, AR/VR wearables). Recent advances such as Cube R-CNN [7] have achieved strong 3D detection performance by training on large, high-quality datasets, while OVMono3D [75] extends this task to the open-vocabulary setting, aiming to detect arbitrary object categories in 3D from monocular views.

Despite progress, a critical bottleneck has been highlighted in recent works [7, 75, 22, 27]: the need for large-scale 3D datasets with high-quality 3D bounding box annotations for effective training and evaluation. This aligns with a broader trend observed in vision and language foundation models [1, 44, 60]: performance improves significantly with more diverse, realistic training data and high-quality supervision. However, unlike 2D image recognition or language tasks, existing 3D datasets [8, 23, 7, 4, 15] are still constrained in scalability, scene diversity, and geometric complexity. As illustrated in Figure 2a, Omni3D [7]—currently the largest public dataset—is dominated by indoor [15, 4, 63, 2, 58] and autonomous driving scenes [24, 9], with limited coverage of common objects (*e.g.*, animals). These constraints hinder both the training and evaluation of generalizable monocular 3D detectors. While prior works [7, 75, 78] demonstrate qualitative success on in-the-wild images from MS-COCO [40], the lack of 3D annotations prevents systematic and quantitative benchmarking. This motivates the central question in this work: *how can we produce high-quality 3D annotations on natural images with minimal human supervision?*

Two major challenges limit scalable 3D dataset construction. First, *collecting 3D data is expensive*: in-the-wild images rarely include depth, and LiDAR or depth sensors require costly hardware and careful calibration, making them far less scalable than RGB capture. Second, *annotating 3D data is labor-intensive*: labeling 3D bounding boxes requires significantly more effort than 2D annotations. Several prior works have attempted to address these issues. For instance, OVM3D-Det [27] lifts 2D images into 3D using off-the-shelf *metric* depth estimation models [53, 6] to generate pseudo-LiDAR data, and infer 3D bounding boxes using object size priors from large language models (*e.g.*, average height for a pedestrian is 1.7 meters). While effective for objects with consistent sizes (*e.g.* cars), this approach struggles with categories exhibiting high intra-class variations (*e.g.*, baby *vs.* adult elephant). It also relies heavily on accurate metric depth prediction—a task that remains fundamentally ill-posed when relying solely on 2D input, as object appearance is entangled with both focal length and actual distance to the camera. Another approach, 3D Copy-Paste [22], augments 3D annotations by inserting synthetic 3D models into images, which however induces the sim-to-real challenge.

To address these limitations, we propose LabelAny3D, an automatic 3D annotation pipeline that efficiently generates 3D bounding boxes for objects across arbitrary categories. Unlike prior works [22, 27], LabelAny3D adopts an *analysis-by-synthesis* paradigm: it reconstructs the holistic 3D scene from monocular images and uses the synthesized representation to infer spatially consistent 3D object annotations (Figure 2b and 3). Our framework is motivated by three key observations: (1) although metric depth estimation remains ill-posed, relative depth estimation [66] is significantly more reliable and consistent; (2) recent advances in object-centric 3D reconstruction, powered by large-scale 3D shape datasets [17] and generative modeling techniques [72], have enabled accurate shape recovery; and (3) 2D vision foundation models [33, 44] offer strong generalization capabilities across diverse, in-the-wild visual domains.

By integrating diverse vision foundation models, LabelAny3D produces high-quality 3D annotations suitable for training monocular 3D models and constructing benchmarks with minimal human supervision. Our experiments demonstrate that the 3D annotations from our pipeline lead to consistent improvements in monocular 3D detection across multiple benchmarks (Figure 2c), outperforming existing auto-labeling methods [27]. We further introduce COCO3D, a new benchmark curated from MS-COCO [40] validation images using our pipeline with human refinement, covering a wide variety of everyday object categories encountered in the wild, as presented in Figure 1.

In summary, our key contributions are as follows:

- We introduce LabelAny3D, an efficient 3D annotation pipeline that generates high-quality 3D bounding boxes on in-the-wild images in an *analysis-by-synthesis* manner.
- We demonstrate that 3D annotations from LabelAny3D consistently improve monocular 3D detection performance, surpassing existing auto-labeling approaches.
- We curate COCO3D, a new benchmark for open-vocabulary monocular 3D detection, featuring a diverse range of object categories beyond those covered in existing 3D datasets.

## 2    Related work

**3D Datasets.**    Many datasets have been developed to support 3D detection.  KITTI [23] and nuScenes [8] focus on autonomous driving, offering LiDAR and camera data for object detection and tracking in urban environments. SUN RGB-D [63], Hypersim [58] and ARKitScenes [4] target indoor settings, capturing room-scale layouts with depth and semantic annotations. Objectron [2] leverage mobile devices to collect real-world 3D object scans, enabling fine-grained object-centric learning. Omni3D [7] unifies multiple datasets to create a large-scale benchmark for general 3D object detection, yet existing benchmarks remain limited in their coverage of diverse, open-world scenarios.  In contrast, we propose to extend 3D datasets beyond indoor and autonomous driving domains. This enables broader generalization across diverse, in-the-wild scenarios.

**Monocular 3D Detection.** Early studies on this task focused predominantly on specialized applications within either outdoor [14, 82, 83, 68, 13, 81, 26, 67] or indoor environments [16, 28, 49, 61, 35], particularly targeting autonomous driving and room layout estimation. The Omni3D dataset facilitated Cube R-CNN [7] in pioneering unified monocular 3D detection.  Subsequently, Uni-MODE [39] extended these advances by proposing the first successful BEV-based detector applicable across diverse environments.  Despite these achievements, most existing methods are constrained by closed vocabularies.  Recent advancements in open-vocabulary 3D detection [46, 47, 11, 12, 77, 84, 80, 79, 52, 20, 69] primarily focus on utilizing 3D point clouds, while OVMono3D [75] first explores the open vocabulary 3D detection task with only monocular image as input. DetAny3D [78] further boosts performance through superior 2D priors and extensive training data.  However, these models still face challenges with generalization to in-the-wild imagery like MS-COCO [40], highlighting the ongoing issue of domain coverage limitations in training datasets.

**Label-efficient 3D Detection.** Due to the high cost of 3D labeling, previous studies have investigated approaches to reduce the reliance on 3D supervision in monocular 3D detection tasks [27, 71, 31, 10, 25]. For instance, Huang *et al.* [27] utilizes open-vocabulary 2D models and pseudo-LiDAR to automatically annotate 3D objects in RGB images, while 3D Copy-Paste [22] inserts synthetic 3D object shapes into 2D images to augment 3D annotations. Additionally, SKD-WM3D [31] introduces a weakly-supervised monocular 3D detection framework by distilling knowledge from pre-trained depth estimation models. While these weakly-supervised methods have proven effective, their application is often limited to indoor or autonomous driving contexts. In this work, we aim to develop novel techniques to broaden the domain coverage of 3D detection models.

**Model-in-the-loop Data Labeling.** Model-in-the-loop data labeling leverages foundation models to enhance and accelerate data annotation.  Prior works have explored this methodology across various tasks. Stereo4D [32] utilizes stereo depth estimation [42] and 2D tracking [19] models to construct 3D trajectory datasets. Cap3D [48] employs vision-language models such as BLIP2 [38], CLIP [54], and GPT-4 [1] to develop a scalable pipeline for captioning 3D assets. DynPose [59] creates dynamic camera pose datasets by integrating advanced tracking [19] and masking [55, 29] models. Unlike these approaches, our work focuses on the task of 3D bounding box annotation from single-view images. By incorporating advanced models, such as monocular depth estimation [66, 6] and image-to-3D reconstruction [72], our pipeline enables accurate and efficient 3D labeling.

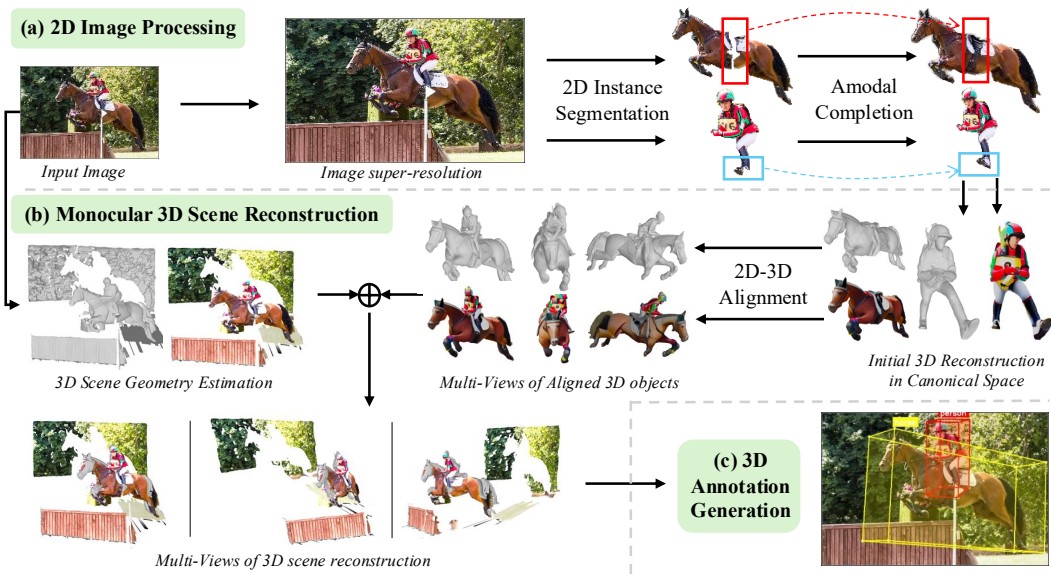

Figure 3: **LabelAny3D**. **(a)** Given an image, we first extract high-resolution object crops; **(b)** A holistic 3D scene is then built from robust depth estimation, 3D object reconstruction, and 2D-3D alignment algorithms. **(c)** Lastly, 3D labels can be easily extracted from the reconstructed 3D scene.

## 3 LabelAny3D: Automatic 3D Labeling via 3D Reconstruction

This section details the proposed LabelAny3D annotation pipeline. As shown in Figure 3, the pipeline generates pseudo annotations from an input image through the following steps.

**Image Super-resolution.** Many objects in MS-COCO [40] appear at low resolution due to factors such as small object scale or compression artifacts, which poses challenges for downstream tasks like 3D reconstruction. To address this, we leverage InvSR [76], a diffusion-based super-resolution (SR) model, to enhance the input image by a factor of $\times 4$. This improves perceptual quality by recovering fine details and sharpening object boundaries. Given an input image $I \in \mathbb{R}^{H \times W \times 3}$, the enhanced image is $I^{\text{SR}} \in \mathbb{R}^{4H \times 4W \times 3}$.

**2D Instance Segmentation.** Prior studies have shown that the ground truth segmentation masks in MS-COCO [40] often exhibit annotation errors. To mitigate this, we leverage the COCONut dataset [18], which provides refined and high-quality segmentation masks built upon the MS-COCO [40] annotations. Given an image and its super-resolved version $I^{\text{SR}}$, we compute the intersection of each object mask $M$ with a boundary mask and if the total intersection of an object exceeds a threshold, the object is considered truncated and excluded. Next, each mask $M$ is upscaled to $M^{\text{SR}} \in \mathbb{R}^{4H \times 4W}$ using nearest-neighbor interpolation to match the resolution of $I^{\text{SR}}$. We also remove any object whose post-processed mask area is below a threshold, as such objects are too small for reliable geometry. Using $I^{\text{SR}}$ and $M^{\text{SR}}$ we extract the enhanced crop of the target object.

**Amodal Completion & 3D Reconstruction.** To handle occluded objects, we adopt an amodal completion strategy inspired by prior work in Gen3DSR [3]. We leverage the learned amodal completion diffusion model from Gen3DSR to inpaint the missing regions of the object crop, generating a completed version $O_{\text{comp}}$ of the targeted object. With $O_{\text{comp}}$ we apply single-view 3D reconstruction methods (*e.g.*, TRELLIS [72]) to recover the full 3D mesh in a canonical pose with normalized scale.

**Scene Geometry Estimation.** We utilize an *affine-invariant* representation of the scene geometry from MoGe [66], and another from a *metric depth estimation model*, such as Depth Pro [6]. To recover the 3D geometry, the MoGe depth map is aligned to the scale and perspective of the metric depth map which is considered to represent real-world geometry [5, 3]. This step ensures that the depth values from MoGe are scaled and transformed to match the metric scale and viewpoint of the target scene. Next, the aligned depth map is *unprojected* into 3D space using the camera intrinsic matrix provided by the MoGe model, to recover the full 3D structure of the target scene.

**Pose Estimation via 2D-3D Alignment.** After obtaining 2D object regions $O_{\text{comp}}$ and 3D reconstructions, we localize the 3D objects within the scene by estimating its pose relative to the input image. This is achieved through dense correspondence matching between the real image and a set of rendered views of the object mesh $\text{Mesh}_{\text{sr}}$. We adopt MASt3R [37] to compute 2D-2D correspondences between the super-resolved real image and the rendered views. Let $x_0 \in \mathbb{R}^2$ denote pixel coordinates in the real image, and $x_1 \in \mathbb{R}^2$ in the rendered view. These matched keypoints are filtered near image borders and invalid depth regions. Using the rendered depth map along with known intrinsics and extrinsics from rendering, we unproject $x_1$ to obtain corresponding 3D points $X_c \in \mathbb{R}^3$ on the $\text{Mesh}_{\text{sr}}$. Given the resulting 3D–2D correspondences $(X_c, x_0)$ and camera intrinsics $K$ inferred from MoGe [66], we apply a Perspective-n-Point (PnP) solver [36] with RANSAC [21] to estimate the relative camera pose $(R, T)$. The object is then transformed into the input image's coordinate frame using this pose, yielding a reconstruction aligned with the relative layout of the original scene.

**Scale Estimation via Depth Alignment.** To recover metric scale, we align the rendered object to the real scene using depth-based scale estimation. Specifically, given the binary mask $M$ from segmentation and the rendered mask $M_{\text{render}}$ obtained using the previously estimated pose, we compute an overlap region $\Omega = M \cap M_{\text{render}}$. Letting $D_{\text{real}}$ and $D_{\text{render}}$ denote the real and rendered depth maps from the same pose, respectively, we estimate the scale factor $s$ as the median depth ratio:

$$s = \text{median}\left(\frac{D_{\text{real}}(\Omega)}{D_{\text{render}}(\Omega)}\right).$$

The estimated scale $s$ is applied to the rotation and translation parameters to form the final transformation matrix. This transformation places the reconstructed 3D object into the metric-scale scene point cloud, completing the 3D scene reconstruction.

**3D Annotation Generation.** In this step, we uniformly sample a point cloud from the mesh surface, capturing the object's geometry in its posed state. As TRELLIS [72] generates objects in a canonical pose with the upright direction aligned to gravity, following prior works [27, 51], we align the vertical axis of the bounding box with this canonical upward direction. To estimate orientation and size, we project the point cloud onto the plane orthogonal to the upward axis and apply PCA to determine the dominant yaw. The point cloud is then rotated to align with the canonical axes, and a tight 3D bounding box is fitted around its extent. This yields the object's 3D bounding box attributes, including center position, orientation, and dimensions.

## 4 Downstream Applications

Section 4.1 introduces a new monocular 3D detection benchmark curated from MS-COCO [40]. Section 4.2 presents our training details using the pseudo labels generated by LabelAny3D.

### 4.1 COCO3D Benchmark

**Human refinement.** We apply LabelAny3D to curate a benchmark for evaluating (open-vocabulary) monocular 3D detection models. To ensure the high quality of our 3D annotations, we employ a human-in-the-loop refinement process. Five annotators with prior 3D vision experience refined the pipeline-generated labels by adjusting bounding box dimensions, rotation, and center position based on the generated point maps. They also had the option to remove any generated bounding box they deemed excessively noisy. In addition, the annotators performed a filtering step to exclude samples containing undesirable characteristics, such as reflections of objects on surfaces like mirrors, windows, or screens, as well as 2D posters or symbols representing 3D objects. Figure 4c presents a selection of diverse examples that were removed during this process. This refinement requires minimal manual effort, supporting efficient large-scale annotation.

**Statistics.** The COCO3D benchmark[2] comprises 2,039 human-refined images with a total of 5,373 instances spanning all 80 categories of the MS-COCO dataset [40]. Figure 1 illustrates samples of the final curated results, along with the categories represented in our COCO3D benchmark. We construct this benchmark using the validation set of the MS-COCO dataset. Figure 4a presents the distribution of the top 50 categories within the COCO3D benchmark. Notably, the *person* category in MS-COCO includes a wide range of examples featuring individuals in various poses, sizes, and

---

[2]A larger version 2 will be released soon. For this camera-ready version, we present results on version 1.

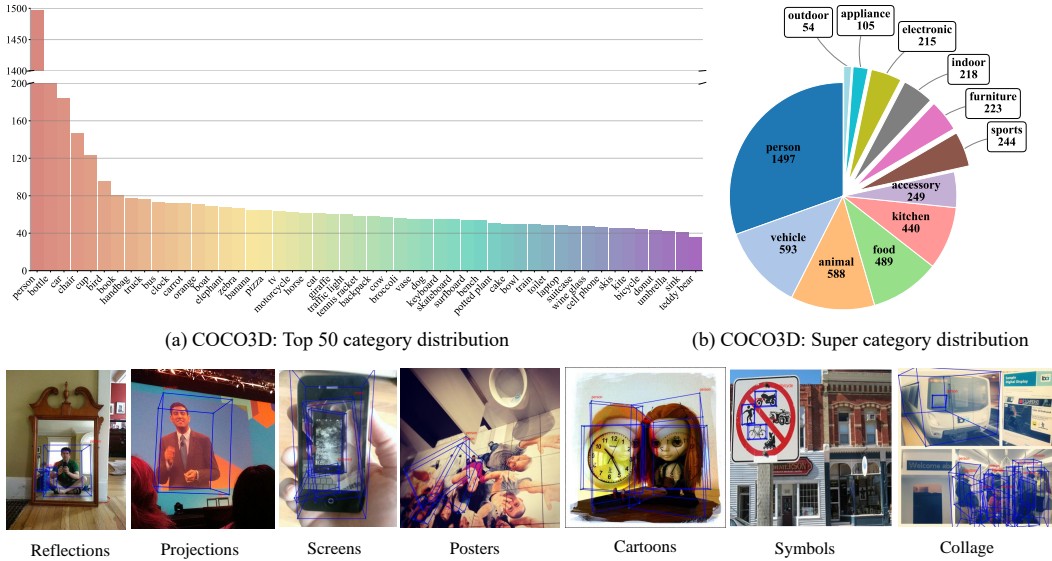

(a) COCO3D: Top 50 category distribution

(b) COCO3D: Super category distribution

(c) COCO3D: Removed photo samples during human refinement based on mentioned characteristics

Figure 4: **COCO3D benchmark. (a)** Distribution of the top 50 categories in the COCO3D benchmark. **(b)** Super-category-wise distribution in the COCO3D benchmark, based on MS-COCO [40]. **(c)** Examples of samples removed from COCO3D during the human refinement process.

age groups, contributing to a rich and diverse set of samples. Figure 4b displays the distribution of supercategories, further demonstrating the diversity captured by our benchmark.

## 4.2 Train Monocular 3D Detector with LabelAny3D

We build our model upon OVMono3D [75], a state-of-the-art open-vocabulary monocular 3D detection model. OVMono3D consists of two stages: (1) detecting and localizing objects in 2D using open-vocabulary detectors (*e.g.*, Grounding DINO [45]); and (2) lifting 2D bounding boxes to 3D cuboids in a class-agnostic manner. Specifically, given an image $I$, a text prompt $T$, and 2D bounding boxes with category labels from a 2D detector, OVMono3D extracts multi-scale feature maps from pretrained vision transformers (*e.g.*, DINOv2 [50]) for each 2D bounding box. These features are then processed by a feed-forward network to predict 3D attributes.

**Training objective.** We closely follow prior work [7] to train our models. We train only the lifting head of OVMono3D [75] using ground-truth 2D bounding boxes. The training objective of is defined as:

$$\mathcal{L} = \sqrt{2}\,\exp(-\mu)\,\mathcal{L}_{\text{3D}} + \mu, \tag{1}$$

where $\mathcal{L}_{\text{3D}}$ is the loss from the 3D cube head, and $\mu$ denotes the uncertainty score. The 3D loss $\mathcal{L}_{\text{3D}}$ consists of disentangled losses for each 3D attribute [62]:

$$\mathcal{L}_{\text{3D}} = \sum_{a} \mathcal{L}_{\text{3D}}^{(a)} + \mathcal{L}_{\text{3D}}^{\text{all}}, \tag{2}$$

where $a \in \{(x_{\text{2D}}, y_{\text{2D}}),\ z,\ (w, h, l),\ r\}$ denotes groups of 3D attributes: 2D center shift, depth, dimensions, and rotation. Each component loss $\mathcal{L}_{\text{3D}}^{(a)}$ isolates the error of a specific attribute group by substituting all other predicted variables with their ground-truth counterparts when constructing the predicted 3D bounding box $B_{\text{3D}}$. The holistic loss $\mathcal{L}_{\text{3D}}^{\text{all}}$ compares the predicted 3D bounding box with the ground truth using the Chamfer Loss:

$$\mathcal{L}_{\text{3D}}^{\text{all}} = \ell_{\text{Chamfer}}(B_{\text{3D}},\ B_{\text{3D}}^{\text{gt}}). \tag{3}$$

**Training Data.** We curate a training set of 15,869 images from the MS-COCO [40] training split, annotated using our LabelAny3D pipeline *without any human refinement*. This pseudo-labeled dataset is used to either train the OVMono3D model from scratch or fine-tune it. For fine-tuning, we

Table 1: **Performance of OVMono3D [75] with different training settings.** We report scores on our COCO3D benchmark, and the novel and base category splits of Omni3D in OVMono3D evaluation. The best results for each metric are highlighted in **bold**. The second best is underlined. "Rel" denotes the relative layout metrics. "*" denotes the model is initialized with the pretrained OVMono3D.

| Training dataset | COCO3D | | | | Omni3D Novel | | Omni3D Base | |
|---|---|---|---|---|---|---|---|---|
| | $AP_{3D}\uparrow$ | $AR_{3D}\uparrow$ | $AP_{3D}^{Rel}\uparrow$ | $AR_{3D}^{Rel}\uparrow$ | $AP_{3D}\uparrow$ | $AR_{3D}\uparrow$ | $AP_{3D}\uparrow$ | $AR_{3D}\uparrow$ |
| Baseline: Omni3D [7, 75] | 5.87 | 10.51 | 20.86 | 30.06 | 16.05 | 36.85 | **24.77** | **47.28** |
| OVM3D-Det* [27] | 2.69 | 5.25 | 7.98 | 12.25 | 5.30 | 15.71 | 7.32 | 26.34 |
| Omni3D [7] + OVM3D-Det [27] | 6.82 | 11.94 | 20.76 | 27.69 | 15.55 | **37.18** | 22.35 | 42.68 |
| LabelAny3D | 7.78 | 15.41 | 24.66 | 34.54 | 8.47 | 23.34 | 3.92 | 19.66 |
| Omni3D [7] + LabelAny3D | **10.92** | **20.10** | **32.02** | **43.82** | **16.98** | 36.96 | 22.74 | 42.46 |

initialize from the OVMono3D model pretrained on Omni3D [7] and further train it on the combined LabelAny3D and Omni3D datasets. For training from scratch, the model is trained solely on the LabelAny3D annotations without relying on any external ground truth 3D supervision.

# 5 Experiments

## 5.1 Experimental Setup

**Benchmarks.** LabelAny3D is evaluated on our COCO3D benchmark. In this benchmark, we exclude 10 of the 80 categories from evaluation due to either too few evaluation instances or extreme aspect ratios (*e.g.*, spoon, baseball bat). We also assess the open-vocabulary detection capabilities of our trained model on Omni3D [7], which primarily encompasses indoor datasets such as SUN RGB-D [63], ARKitScenes [4], and Hypersim [58]; the object-centric dataset Objectron [2]; and autonomous driving datasets including nuScenes [8] and KITTI [24].

**Baselines.** We evaluate LabelAny3D in terms of both pseudo annotation quality and its effectiveness for training open-vocabulary monocular 3D detectors. Specifically, we compare LabelAny3D with OVM3D-Det [27] on annotation quality and downstream detection performance. Additionally, using our curated COCO3D validation set, we benchmark the performance of OVMono3D [75] and OVMono3D fine-tuned on our pseudo annotations derived from the MS-COCO [40] training set.

**Evaluations.** Following [7, 75], we report mean $AP_{3D}$ and $AR_{3D}$, computed using Intersection-over-Union (IoU) between predicted and ground-truth 3D bounding boxes across IoU thresholds ranging from 0.05 to 0.50 in increments of 0.05.

Since the metric depth in our COCO3D validation set is derived from model predictions, it may contain biases due to the inherent difficulty in accurately predicting and human-validating absolute depth. In contrast, the relative depth from MoGe [66] is rigorously verified through human refinement, providing greater reliability. To address this, we introduce a novel metric, *Relative Layout $AP_{3D}$*, within our COCO3D benchmark. This metric assesses the consistency of the relative spatial layout between predicted and ground-truth bounding boxes, in the same spirit with the relative depth evaluation [73, 74]. Specifically, we align predicted bounding boxes $\hat{B}$ and ground-truth bounding boxes $B$ by optimizing a global scale factor $s \in \mathbb{R}^+$:

$$s^* = \arg\max_s \frac{1}{N} \sum_{i=1}^{N} \text{IoU}_{3D}(s \cdot \hat{B}_i, B_i),$$

where $N$ is the total number of boxes. Due to the non-differentiability of $\text{IoU}_{3D}$ with respect to $s$, we perform a grid search within a bounded interval. $AP_{3D}$ and $AR_{3D}$ are then computed on the aligned, scale-normalized predictions, emphasizing relative 3D box layout rather than metric accuracy.

## 5.2 Main Results

**LabelAny3D improves monocular 3D detection on COCO3D.** Table 1 benchmarks OV-Mono3D [75] trained with different datasets. When pretrained solely on Omni3D [7], the model exhibits limited generalization to COCO3D images, likely due to substantial domain gaps. To improve performance, we generate pseudo labels on the MS-COCO [40] training set using the auto-labeling

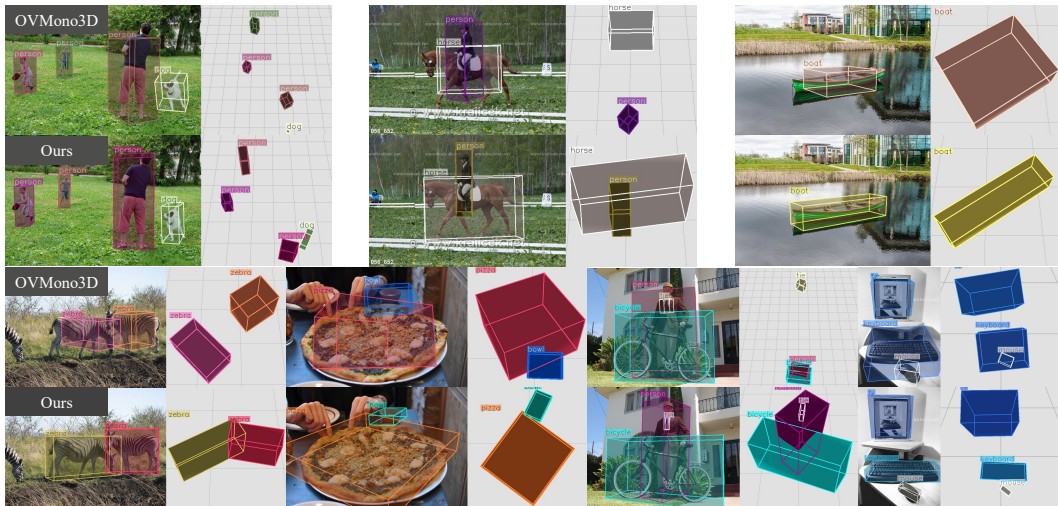

Figure 5: Qualitative open-vocabulary 3D detection results on in-the-wild-images: OVMono3D [75] *vs.* our fine-tuned OVMono3D. We display both the 3D predictions overlaid on the image and a top-down view with a base grid of $1\,\mathrm{m} \times 1\,\mathrm{m}$ tiles.

Table 2: Pseudo annotation quality on COCO3D benchmark. The best results for each metric are highlighted in **bold**. "Rel" denotes the relative layout metrics. For fair comparison, we use the same depth for OVM3D-Det as for ours, denoted by "*".

| Methods | $AP_{3D}\uparrow$ | $AP_{3D}^{15}\uparrow$ | $AP_{3D}^{25}\uparrow$ | $AP_{3D}^{50}\uparrow$ | $AR_{3D}\uparrow$ | $AP_{3D}^{Rel}\uparrow$ | $AR_{3D}^{Rel}\uparrow$ |
|---|---|---|---|---|---|---|---|
| OVM3D-Det* [27] | 10.03 | 16.88 | 9.03 | 1.44 | 17.82 | 10.04 | 17.84 |
| LabelAny3D | **64.17** | **82.11** | **74.47** | **57.34** | **73.57** | **64.17** | **73.57** |

pipeline proposed in OVM3D-Det [27]. However, training OVMono3D from scratch with these labels fails to converge. Even when initialized from the pretrained OVMono3D, fine-tuning on OVM3D-Det labels alone leads to poor performance. When fine-tuned on the combined dataset of Omni3D and OVM3D-Det, the model achieves only a marginal 0.95 $AP_{3D}$ gain on COCO3D and shows degraded performance on novel categories, suggesting that excessive label noise negatively impacts learning.

In contrast, training OVMono3D from scratch on pseudo-labels generated by our LabelAny3D exhibits better performance on COCO3D, demonstrating the effectiveness of our pipeline in supporting model training. Notably, the model trained solely on COCO3D pseudo labels achieves 8.47 $AP_{3D}$ on OVMono3D's out-of-domain novel categories, highlighting its improved generalizability. Further gains are observed when fine-tuning the pretrained OVMono3D model on the combined Omni3D and LabelAny3D pseudo-labeled datasets, resulting in a 5.05 $AP_{3D}$ increase on COCO3D and improved performance across novel categories. These results validate the effectiveness of our LabelAny3D pipeline in producing high-quality, in-the-wild 3D annotations.

Despite these gains, all fine-tuned models show some degradation on OVMono3D's base categories. We attribute this to two factors: (1) increased scene and category diversity without a corresponding increase in model capacity, leading to catastrophic forgetting; and (2) label noise in the pseudo annotations, which may introduce harmful gradients during training.

Figure 5 presents qualitative examples on COCO3D. Compared to the baseline model, our fine-tuned OVMono3D demonstrates stronger detection robustness in diverse scenes, particularly for novel object categories that are underrepresented in existing datasets, such as *animals*, *pizza*, *tie*, *boat*.

**LabelAny3D achieves better pseudo annotations.** Table 2 compares LabelAny3D with OVM3D-Det [27] on the quality of pseudo annotations. LabelAny3D consistently outperforms the baseline across all reported metrics. The $AP_{3D}$ between automatically generated and human-refined annotations is 64.17, indicating minimal manual correction and demonstrating the efficiency of our pipeline. Figure 6 provides a visual comparison with OVM3D-Det [27], revealing the limitations of metric-based priors in in-the-wild settings, where object size and shape vary widely – even within the same category (*e.g.* baby elephants, children, and boats). In such cases, metric depth estimates

| Input Image | OVM3D-Det | LabelAny3D | Input Image | OVM3D-Det | LabelAny3D |

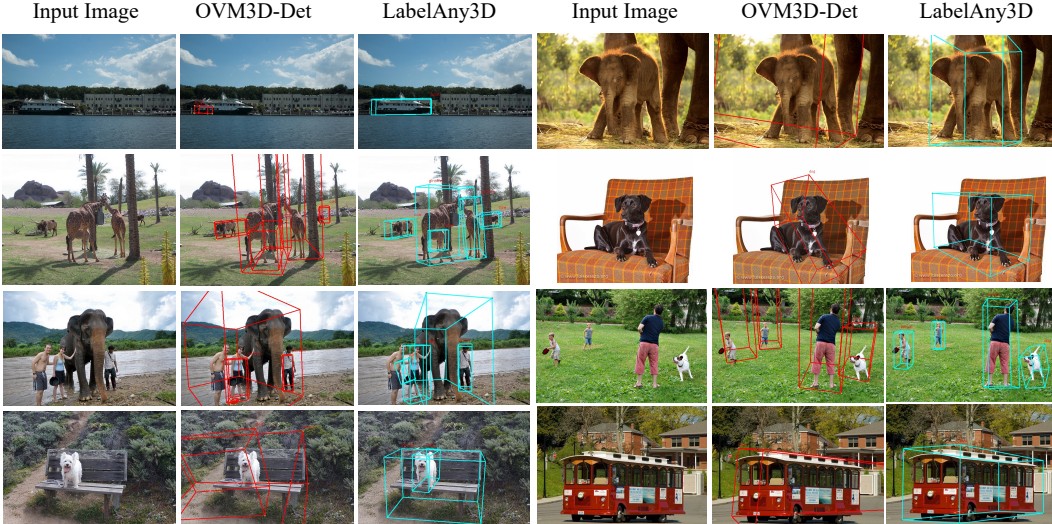

Figure 6: Qualitative comparisons between OVM3D-Det [27] and LabelAny3D *without any human refinement.* These examples illustrate that OVM3D-Det often produces inaccurate metric dimensions for categories with high intra-class size variability, such as humans, animals, vehicles, and furniture.

prone to inaccuracy, leading to misaligned results. In contrast, LabelAny3D leverages relative depth and mesh reconstruction to produce appearance-consistent 3D bounding boxes, resulting in higher annotation fidelity. We further compare the quality of pseudo annotations produced by LabelAny3D and OVM3D-Det [27] on the KITTI [24] benchmark. LabelAny3D achieves a higher overall $AP_{3D}$ of 13.6, compared to 12.39 from OVM3D-Det. Notably, on the truck category, LabelAny3D significantly outperforms OVM3D-Det with an $AP_{3D}$ of 32.74 vs. 13.46, highlighting its effectiveness.

**Ablation of components in LabelAny3D.** We conduct ablation studies on a subset of the COCO3D dataset and report annotation quality in Table 3. The vanilla 3D scene reconstruction model Gen3DSR [3] achieves an $AP_{3D}$ of only 1.95. In contrast, our full LabelAny3D pipeline achieves a higher $AP_{3D}$ of 43.17. This results shows our pipeline provides better 3D reconstruction for 3D box labeling. Removing the image super-resolution module leads to a substantial drop to 28.13 $AP_{3D}$, highlighting its importance in enhancing detail for small and distant objects. Eliminating the amodal completion also reduces performance, showing its role to alleviate the negative impact of occlusion.

Table 3: Ablation studies. Default settings are marked in gray .

| Framework | $AP_{3D}$ ↑ |
|---|---|
| Gen3DSR [3] | 1.95 |
| LabelAny3D | **43.17** |
| – w/o Super Resolution | 28.13 |
| – w/o Amodal Completion | 39.22 |
| – w/o MoGe [66] | 22.77 |
| – w/ DreamGaussian [64] | 36.84 |
| – w/ ICP | 24.28 |

Replacing MoGe's [66] relative depth (scaled to match Depth Pro [6]) with Depth Pro alone leads to a significant drop in performance, showing that MoGe provides more accurate and reliable relative depth. For 3D reconstruction, using TRELLIS [72] improves $AP_{3D}$ by 6.33 points over DreamGaussian [64], suggesting TRELLIS produces more realistic and higher-fidelity reconstructions. To align reconstructed objects with the image point cloud, we compare iterative closest point (ICP) against our used 2D matching + Perspective-n-Point (PnP) method. The latter yields superior results, largely due to the robustness of the underlying 2D matching model.

## 6   Discussion

In this work, we introduced LabelAny3D, an *analysis-by-synthesis* pipeline for annotating 3D bounding boxes of arbitrary objects from monocular, in-the-wild images. Our pipeline integrates specialized vision foundation models to reconstruct 3D scenes and derive accurate 3D annotations. Leveraging LabelAny3D, we curated COCO3D, a new 3D detection benchmark encompassing

diverse object categories beyond existing datasets, with minimal human intervention. Our findings demonstrate that LabelAny3D effectively enhances open-vocabulary 3D detection performance with minimal human effort and enabling the large-scale development of diverse 3D datasets. Our pipeline also has the potential to benefit other 3D scene understanding tasks, such as amodal 3D reconstruction, 6D pose estimation, and scene completion. See supplementary material for implementation details, more qualitative results and analysis.

## 7 Limitations

While our LabelAny3D pipeline leverages mature foundation models for depth estimation, camera intrinsic estimation, amodal completion, image-to-3D generation, and matching, these models can still fail in challenging scenarios involving heavy occlusion, textureless regions, or small objects, which introduce noise into the final 3D bounding box annotations.

The RGB-to-3D model TRELLIS [72] in our pipeline may generate meshes with ambiguous depth along the viewing direction, causing misalignment with RGBD point clouds and inaccurate bounding boxes. Future work could condition 3D generation on RGBD data, as in Hunyuan3D-Omni [65].

Additionally, future work could investigate robust training strategies for noisy pseudo-annotations.

To ensure benchmark reliability, we exclude objects with erroneous depth estimation, severe occlusion, or truncation. Consequently, our dataset is not exhaustively annotated but can evaluate 2D box-prompted methods like OVMono3D [75] and DetAny3D [78], or end-to-end 3D detection methods using proximity-based metrics.

Since our pipeline integrates depth estimation and amodal completion as modular APIs, future improvements in these components can be directly incorporated to enhance annotation quality. These limitations underscore the need for more robust auto-labeling frameworks and training strategies for open-vocabulary monocular 3D detection.

## 8 Acknowledgement

The authors acknowledge the University of Virginia Research Computing and Data Analytics Center, Advanced Micro Devices AI and HPC Cluster Program, Advanced Cyberinfrastructure Coordination Ecosystem: Services & Support (ACCESS) program, and National Artificial Intelligence Research Resource (NAIRR) Pilot for computational resources, including the Anvil supercomputer (National Science Foundation award OAC 2005632) at Purdue University and the Delta and DeltaAI advanced computing resources (National Science Foundation award OAC 2005572). This work was supported by Adobe Research Gift, the National Science Foundation (awards 2129824, 2312487, and 2403060), United States Army Research Office (grant W911NF-24-1-0089), and Lockheed Martin Advanced Technology Labs. The authors thank Hao Gu, Guangyi Xu, and Jiahui Zhang for annotation assistance.

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

# Appendix

## A   Human Refinement Interface

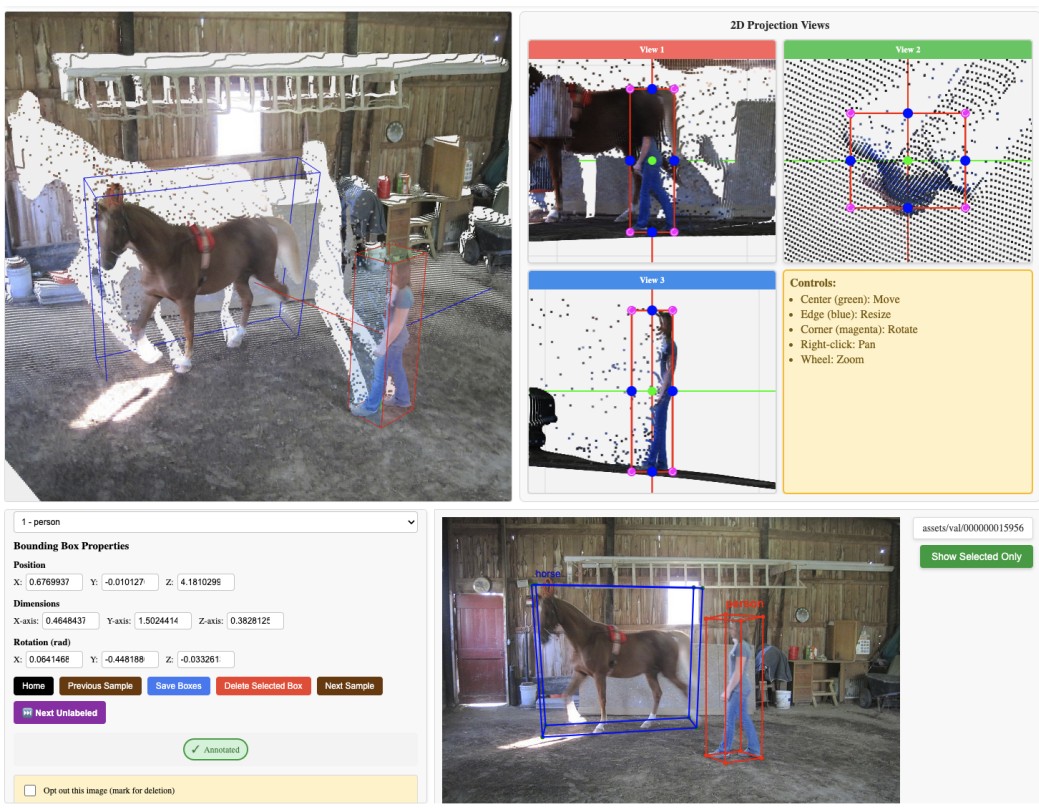

Figure 7: Human refinement interface.

Figure 7 illustrates our annotation interface for human refinement of LabelAny3D's automatically generated pseudo labels on COCO3D. The top left panel displays the point cloud and corresponding 3D bounding boxes, which annotators can rotate and translate for an optimal viewing angle. The top-right panel shows three 2D projection views of the point cloud along the local axes of the 3D bounding box. Annotators can adjust the 3D bounding box by manipulating the edges, corners, or center in each view. The lower-left panel shows the attributes of the selected bounding box. Annotators can modify specific parameters (e.g., width) using keyboard shortcuts. They also have the option to delete a bounding box if it corresponds to an invalid 3D object (e.g., a person on a poster). The lower-right panel visualizes the 2D projections of the current 3D boxes to provide additional context. Annotators may also choose to discard the entire image if the relative geometry is incorrect or no valid 3D objects are present.

## B   Annotation Efficiency

Table 4 presents a category-wise overview of $AP_{3D}$, $AR_{3D}$, and average $IoU_{3D}$ for the pseudo annotations generated by LabelAny3D, evaluated against our refined COCO3D benchmark. The results demonstrate that our pipeline produces high-quality annotations with human-like accuracy. In addition to achieving high precision, the average IoU exceeds 0.40 for the majority of categories, indicating strong alignment of spatial layout.

Table 2 compares the overall $AP_{3D}$ of our method against the baseline OVM3D-DET [27], with our approach achieving a higher AP of 64.17. Among the 5,373 annotations in the COCO3D benchmark, 3,146 were accepted by human annotators without modification, while 2,227 required only minor refinement. Just 466 were rejected due to issues such as reflections, 2D object representations, or insufficient point cloud quality.

These results demonstrate that LabelAny3D produces high-quality pseudo annotations that can serve as effective initialization for human annotators. With minimal effort required for refinement, our pipeline significantly reduces manual workload and accelerates the overall annotation process.

Table 4: **Per-category 3D annotation quality for the top 50 categories**. Comparison between pseudo annotations from LabelAny3D and human-refined annotations. Ranking is based on $IoU_{3D}$.

| Category | $AP_{3D}\uparrow$ | $AR_{3D}\uparrow$ | $IoU_{3D}\uparrow$ | Category | $AP_{3D}\uparrow$ | $AR_{3D}\uparrow$ | $IoU_{3D}\uparrow$ |
|---|---|---|---|---|---|---|---|
| sports ball | 81.58 | 88.26 | 80.94 | bed | 88.84 | 93.33 | 76.32 |
| fire hydrant | 91.14 | 94.80 | 75.16 | airplane | 90.61 | 93.53 | 68.83 |
| couch | 70.05 | 88.57 | 67.89 | snowboard | 63.50 | 72.14 | 68.27 |
| parking meter | 92.56 | 94.29 | 63.94 | mouse | 68.98 | 78.70 | 63.57 |
| vase | 89.89 | 92.73 | 61.38 | skateboard | 69.87 | 80.73 | 60.70 |
| fork | 64.76 | 75.65 | 58.93 | cat | 72.94 | 80.49 | 57.66 |
| bicycle | 83.45 | 94.65 | 56.60 | microwave | 82.65 | 90.53 | 56.00 |
| dog | 79.54 | 90.36 | 54.73 | tie | 48.71 | 67.83 | 54.60 |
| bus | 84.28 | 93.61 | 52.20 | dining table | 86.38 | 92.22 | 51.59 |
| bench | 84.80 | 89.81 | 49.43 | surfboard | 50.10 | 65.00 | 48.24 |
| sink | 60.22 | 71.95 | 46.97 | sandwich | 65.13 | 77.94 | 47.38 |
| refrigerator | 72.87 | 86.52 | 45.04 | kite | 70.54 | 80.71 | 44.99 |
| laptop | 69.56 | 84.69 | 43.92 | cake | 80.60 | 90.40 | 43.38 |
| oven | 66.77 | 79.05 | 43.29 | umbrella | 92.28 | 95.21 | 41.30 |
| tv | 43.71 | 62.54 | 41.10 | bowl | 79.32 | 88.40 | 40.70 |
| horse | 76.75 | 89.84 | 40.38 | bear | 82.62 | 89.71 | 40.28 |
| backpack | 69.39 | 82.11 | 38.90 | keyboard | 47.08 | 61.82 | 38.36 |
| pizza | 70.90 | 81.82 | 37.22 | skis | 40.44 | 64.25 | 36.93 |
| traffic light | 62.11 | 77.12 | 36.80 | cup | 89.94 | 93.44 | 36.35 |
| remote | 59.63 | 71.60 | 35.49 | tennis racket | 38.10 | 59.31 | 31.62 |
| clock | 39.06 | 57.36 | 33.25 | elephant | 87.71 | 98.82 | 33.19 |
| handbag | 68.75 | 82.28 | 32.23 | wine glass | 97.11 | 98.51 | 31.76 |
| chair | 61.75 | 88.63 | 32.60 | horse | 76.75 | 89.84 | 40.38 |
| boat | 80.06 | 90.29 | 12.15 | bird | 70.48 | 85.53 | 16.68 |
| motorcycle | 92.65 | 96.77 | 13.99 | book | 65.94 | 76.75 | 13.57 |

## C   Implementation Details

During annotation, we exclude objects whose masks contain fewer than 400 pixels. Following [30], we also discard objects whose masks overlap the image boundary by more than 10 pixels, treating them as truncated.

For depth estimation, we align the scale-invariant depth map from MoGe [66] with the metric-scale depth map predicted by Depth Pro [6] using a global scale transformation. Specifically, we first use MoGe to generate a relative depth map and extract camera intrinsics. Then, we apply Depth Pro to predict a metric-scale depth map for the same scene. We perform RANSAC-based linear regression to fit a robust scale factor that maps MoGe's relative depths to Depth Pro's metric scale. This allows us to retain the fine geometric details from MoGe while calibrating the scale to real-world distances.

For 2D–3D matching, we first estimate the camera elevation angle using the elevation module from One-2-3-45 [43], based on the amodal-completed object crop. We then render 8 views of the object at the estimated elevation, with azimuths spaced at 45-degree intervals. After computing correspondences between the amodal crop and the 8 rendered views, we obtain an initial camera pose.

Using this pose, we render the mesh again and perform an additional 2D–3D matching step to refine the camera pose. Generating pseudo annotations for a single object takes approximately one minute.

Our implementation is based on PyTorch3D [56] and Detectron2 [70]. Following [75], we use DINOv2-Base [50] as the image feature encoder and freeze its parameters during training. The model is initialized from the publicly released OVMono3D weights and fine-tuned for 58k steps with a batch size of 64. We train the model using SGD with an initial learning rate of 0.0012, which decays by a factor of 10 at 60% and 80% of training. A linear warm-up is applied for the first 1.8k steps. Training takes approximately 48 hours on 4 NVIDIA A40 GPUs. We apply standard image augmentations during training, including random horizontal flipping and resizing. In addition, a random positional perturbation is applied to the input 2D bounding box, with a maximum offset ratio of 0.2. For evaluation on COCO3D, we use ground-truth 2D boxes as input. For Omni3D [7], we adopt the same 2D detections from Grounding DINO [45] as used in OVMono3D [75].

## D    Labeling Performance on Additional Benchmarks

We compare LabelAny3D with baseline OVM3D-Det [27] on three established benchmarks: SUN-RGBD [63], nuScenes [9], and Objectron [2], covering indoor, self-driving, and object-centric domains, respectively. We randomly sample approximately 300 images from each dataset's test split in Omni3D [7]. Using ground truth 2D boxes as input, we query SAM [34] to obtain instance masks, then generate 3D boxes using both methods.

Since our pipeline filters out small or heavily occluded objects—as TRELLIS [72], the state-of-the-art 3D reconstruction model used in our pipeline, is sensitive to occlusion and object size—we first evaluate performance on highly visible objects. LabelAny3D achieves $AP_{3D}$ of 37.71, 13.59, and 6.07 on SUN-RGBD [63], nuScenes [9], and Objectron [2], respectively, while baseline OVM3D-Det [27] achieves 37.58, 11.38, and 3.84.

On low-visibility objects, LabelAny3D achieves $AP_{3D}$ of 20.20, 6.69, and 2.54, while OVM3D-Det achieves 25.56, 8.50, and 3.85 on the same benchmarks. These results demonstrate that LabelAny3D performs better on low-occlusion objects, as our 3D scene reconstruction yields more accurate boxes. The baseline method excels on low-visibility objects due to its use of metric-scale object size priors. Note that both methods achieve lower performance on Objectron, likely due to inaccurate metric depth estimation for this dataset.

We further ensemble the two methods, using the baseline for low-visibility objects and LabelAny3D for high-visibility objects. The ensembled method achieves $AP_{3D}$ of 30.23, 9.12, and 4.74 on the three benchmarks, compared to 29.89, 8.22, and 3.73 for the baseline alone, demonstrating that the two methods are complementary.

Table 5: **Comparison of COCO3D with Other Benchmarks.** We report a statistical comparison between COCO3D and existing 3D detection benchmarks on their respective test sets. We report the number of images, categories, covered domains, and the distribution of instances across MS-COCO super categories (percentages reported).

| Benchmark | # Img | # Cat | Domain | Person | Food | Animal | Vehicle | Kitchen | Furniture | Accessory | Indoor | Sports | Electronic | Outdoor | Appliance |
|---|---|---|---|---|---|---|---|---|---|---|---|---|---|---|---|
| SUN RGB-D | 5,050 | 82 | Indoor | 0.1 | - | - | 0.2 | 6 | 65.5 | 8.0 | 11.8 | - | 7.8 | - | 0.5 |
| ARKitScenes | 7,610 | 15 | Indoor | - | - | - | - | 10.3 | 80.5 | - | 3.1 | - | 4.1 | - | 2.1 |
| Hypersim | 7,690 | 29 | Indoor | - | - | - | - | 0.9 | 37.2 | 3.6 | 49.7 | - | 8.5 | - | - |
| Objectron | 9,314 | 9 | Object | - | - | - | 3.4 | 32.4 | 12.5 | 20.9 | 12.8 | - | 17.9 | - | - |
| KITTI | 3,769 | 8 | Driving | 12.9 | - | - | 64.4 | - | - | - | - | - | - | 22.8 | - |
| nuScenes | 6,019 | 9 | Driving | 15.8 | - | - | 63.1 | - | - | - | - | - | - | 21.1 | - |
| COCO3D | 2,039 | 80 | In the Wild | 27.9 | 9.1 | 10.9 | 11.0 | 8.2 | 4.2 | 4.6 | 4.1 | 4.5 | 4.0 | 1.0 | 2.0 |

## E    Comparison of COCO3D with Existing Benchmarks

Table 5 shows a statistical comparison between COCO3D and existing 3D detection benchmarks on their respective test sets. We report the number of images, categories, covered domains, and the distribution of instances across MS-COCO super categories.

Compared to prior benchmarks—which often focus on specific domains such as indoor environments (e.g., SUN RGB-D), object-centric scenes (e.g., Objectron), or self-driving datasets (e.g., KITTI, nuScenes)—COCO3D offers broader coverage across indoor and outdoor everyday scenes. From the

super-category perspective, COCO3D uniquely includes categories from *food*, *animal*, and *sports*, which are often absent in existing 3D datasets. In conclusion, our COCO3D benchmark offers a diverse, real-world, and scalable benchmark for advancing the open-vocabulary monocular 3D detection field.

## F  COCO3D Benchmark Samples

Figure 8 presents additional samples from our human-refined COCO3D benchmark. The results demonstrate that the relative geometry and spatial layout of the scenes and 3D bounding boxes are highly accurate and align well with human perception.

## G  More Qualitative Comparisons

We report more qualitative comparisons of OVMono3D [75] with our finetuned variant in Figure 9. Trained with additional pseudo-labeled in-the-wild images, our model produces more accurate predictions for challenging object categories such as animals, athletes, and food.

## H  Failure Case

Figure 10 illustrates several failure cases of LabelAny3D. In highly occluded scenes, such as the cow in Figure 10(a), the amodal completion model fails to reconstruct the full object, resulting in a 3D bounding box that captures only a partial region. As our method relies on ground-truth 2D instance segmentations, it may incorrectly generate 3D boxes for objects that are not physically present in the 3D scene—for example, the person on a television screen in Figure 10(b). Additionally, for crowded scenes, such as in Figure 10(c), the COCO [40, 18] dataset often lacks per-instance segmentation labels. In such cases, instance segmentation models like Grounded SAM [57] also struggle to separate individual objects accurately, causing our method to miss multiple instances.

## I  Broader Impact

Our work facilitates efficient 3D annotation of objects from any category in diverse, in-the-wild scenes. By integrating the generated pseudo labels, existing open-vocabulary monocular 3D detectors become more robust to out-of-domain categories (e.g., animals), which can enhance the reliability of autonomous systems such as robots and self-driving vehicles, particularly in safety-critical scenarios.

Our dataset is curated from publicly available sources, and therefore does not raise privacy concerns. While our algorithm is category-agnostic and does not introduce explicit bias, the underlying datasets may reflect societal or geographic biases present in the source data. We encourage future work to investigate and mitigate such biases when deploying systems trained on our annotations.

## J  Licenses

Table 6: Licenses of assets used.

| Asset | License |
| --- | --- |
| Cube R-CNN [7] | CC-BY-NC 4.0 |
| Grounding DINO [45] | Apache License 2.0 |
| Segment Anything [34] | Apache License 2.0 |
| Unidepth [53] | CC-BY-NC 4.0 |
| MoGe [66] | MIT License |
| Depth Pro [6] | Apple License (Link) |
| InvSR [76] | S-Lab License 1.0 (Link) |
| One-2-3-45 [43] | Apache License 2.0 |
| TRELLIS [72] | MIT License |
| Gen3DSR [3] | CC-BY 4.0 |
| MASt3R [37] | CC-BY-NC-SA 4.0 |
| OVMono3D [75] | Apache License 2.0 |
| OVM3D-Det [27] | Apache License 2.0 |
| KITTI [24] | CC-BY-NC-SA 3.0 DEED |
| nuScenes [8] | CC-BY-NC 4.0 |
| SUN RGB-D [63] | MIT License |
| ARKitScenes [4] | Apple License (Link) |
| COCO [41] | CC-BY 4.0 |
| COCONut [18] | Apache License 2.0 |

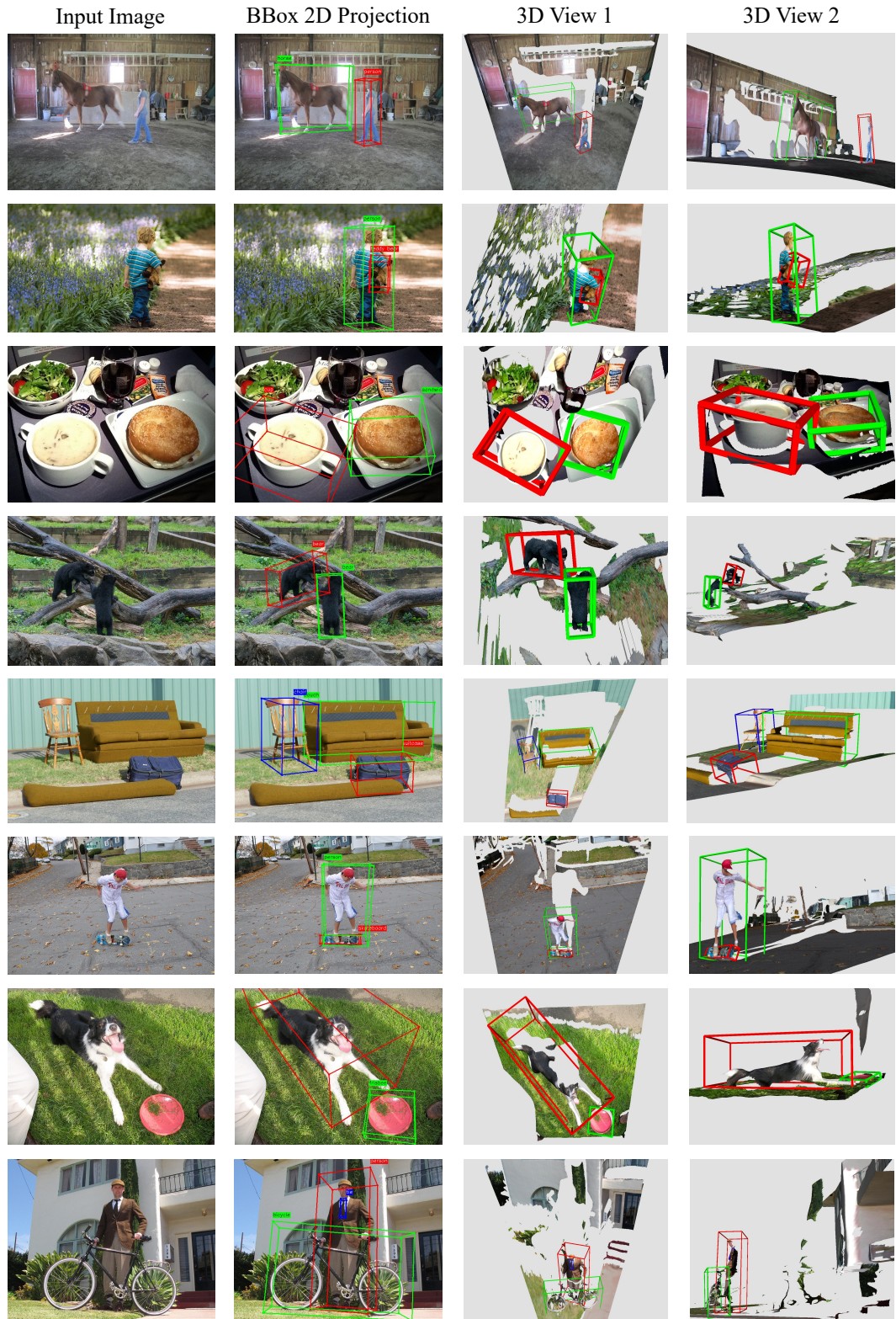

| Input Image | BBox 2D Projection | 3D View 1 | 3D View 2 |

Figure 8: More COCO3D benchmark samples (after human refinement). For each example, we show: (1) the input image, (2) the projected 3D bounding boxes overlaid on the image, and (3–4) two 3D views of the scene point map with the 3D bounding boxes. Please see our project page for rendered videos from 3D Scene and bounding boxes.

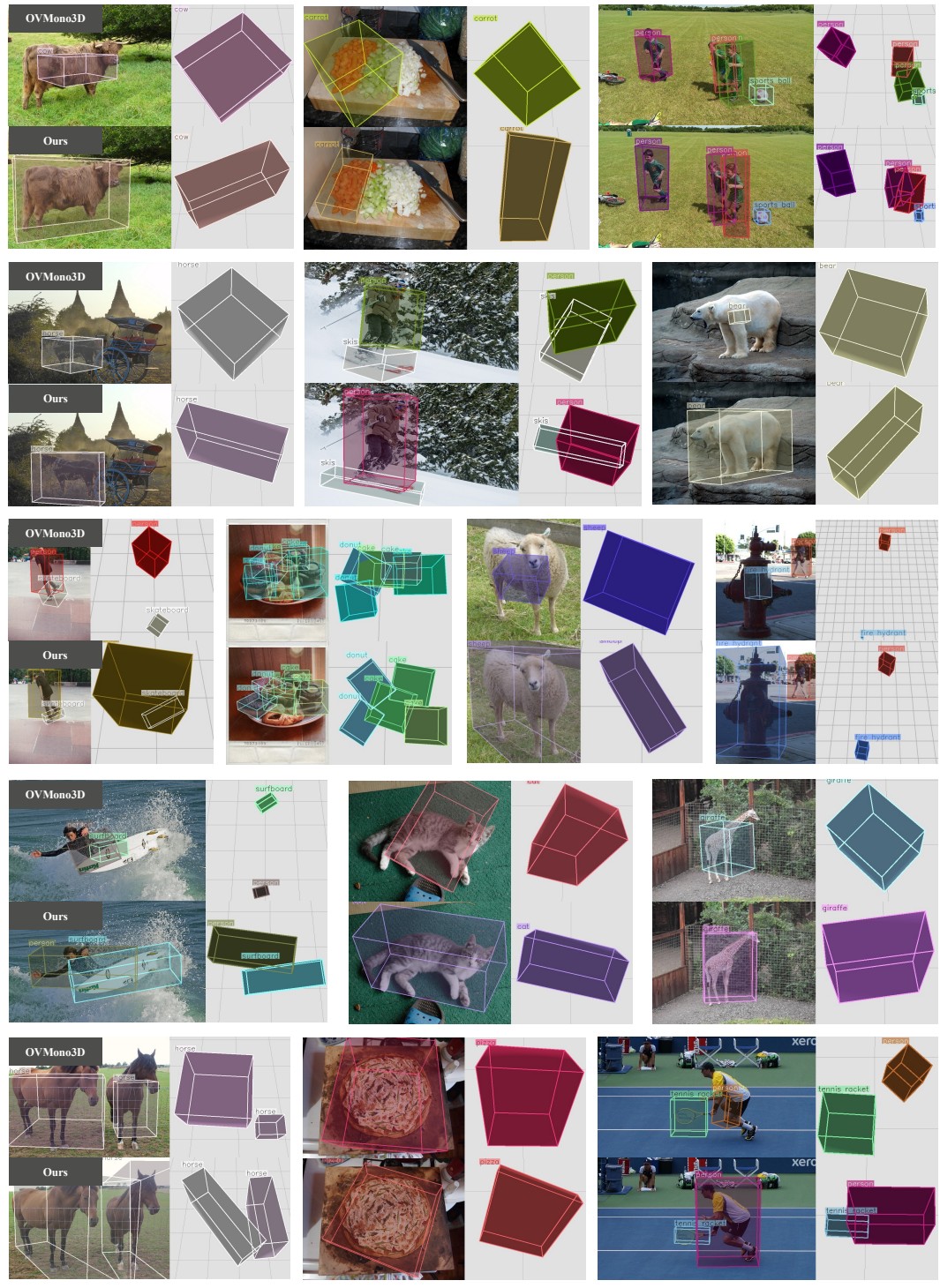

Figure 9: More qualitative open-vocabulary 3D detection results on in-the-wild-images: OV-Mono3D [75] *vs.* our finetuned OVMono3D. We display both the 3D predictions overlaid on the image and a top-down view with a base grid of $1\,\mathrm{m} \times 1\,\mathrm{m}$ tiles.

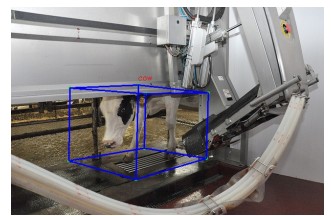 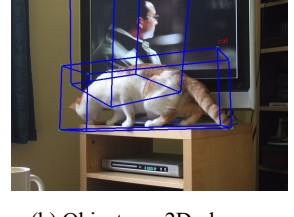

(a) Highly occluded objects        (b) Objects on 2D plane

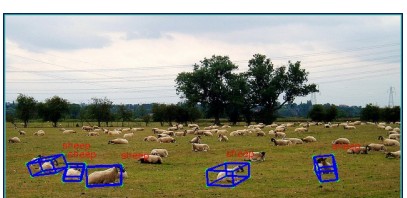 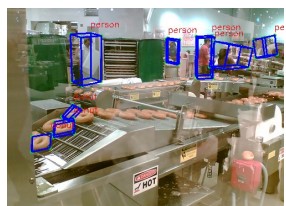

(c) Too crowded objects

Figure 10: Failure cases.

