# OpenReview forum: "LabelAny3D: Label Any Object 3D in the Wild"
_NeurIPS.cc/2025/Conference — NeurIPS 2025 poster_

### Official Review · Reviewer_FwMH · 2025-06-23

**Clarity:** 3
**Significance:** 3
**Originality:** 3
**Rating:** 4
**Confidence:** 4

**Summary:**

This paper presents a pipeline to annotate 3D objects from in the wild images. It is an analysis-by-synthesis framework that reconstructs holistic 3D scenes from 2D images to efficiently produce high-quality 3D bounding box annotations. Built on this pipeline, they present COCO3D, a new benchmark for open-vocabulary monocular 3D detection, derived from the MS-COCO dataset and covering a wide range of object categories absent from existing 3D datasets. Experiments show that annotations generated by LabelAny3D improve monocular 3D detection performance across multiple benchmarks, outperforming prior auto-labeling approaches in quality.

**Questions:**

1. I am confused how to combine MoGe with Depth Pro at Line 141? Why using metric depth estimator there (the paper claim only using relative depth estimator)?

**Ethical Concerns:**

["NO or VERY MINOR ethics concerns only"]

**Final Justification:**

The rebuttal is good. I am willing to accept this paper.

**Limitations:**

1. The paper should test more models to show the effectivess of their new dataset. Now they only test one model, OVMono3D, which I think it is not enough.
2. Why reconstructs the whole 3D scene? I think only reconstrucing the 3D objects are enough to annotate the 3D object bbox.
3. Can you compare the COCO3D benchmark with other open-vocabulary 3D detection benchmarks?

**Quality:**

3

**Strengths And Weaknesses:**

strengths:
1. clear writing and figures.
2. It provides a novel and effective pipeline to annotate 3D objects from any images.
3. A new open-vocabulary 3D detection benchmark derived from MS-COCO.

weakess:
1. The paper should test more models to show the effectivess of their new dataset. Now they only test one model, OVMono3D, which I think it is not enough.
2. Why reconstructs the whole 3D scene? I think only reconstrucing the 3D objects are enough to annotate the 3D object bbox.
3. Can you compare the COCO3D benchmark with other open-vocabulary 3D detection benchmarks?

---

> ### Author Rebuttal · Authors · 2025-07-31
>
> We thank the reviewer for the constructive comments and for recognizing our “clear writing” and “novel and effective pipeline”.
>
> **\[W1\] The paper should test more models to show the effectiveness of their new dataset.**
>
> Thank you for the constructive suggestion\! Open-vocabulary monocular 3D object detection is still an emerging task. To our knowledge, only three works focus specifically on this problem: **OVMono3D** (which we use as our base model), **OVM3D-Det** (our baseline with a pseudo-annotation pipeline), and **DetAny3D** \[1\], recently accepted to ICCV 2025\. DetAny3D’s model weights were released after the NeurIPS submission deadline (on June 25\) and we have incorporated all its baseline comparisons. Due to time and resource  constraints, we will include results from DetAny3D finetuned on our pseudo labels in the revised version. Since the author of DetAny3D didn’t provide evaluation scripts, it needs some engineering effort to complete. We will strive to provide the evaluation results during the discussion period.
>
> **\[W2\] Why reconstructs the whole 3D scene? Only reconstructing the 3D objects are enough to annotate the 3D object bbox.**
>
> Great question\! While we use TRELLIS for 3D object reconstruction, it generates objects in a normalized canonical space, without alignment to image pixels or the global scene. To obtain accurate 3D bounding box attributes—such as center, size, and pose—we need to map these objects back into their true positions within the scene. To achieve this, we first reconstruct a scene-level point map, and then perform 2D–3D alignment using MASt3R and a PnP solver. This alignment step is essential to recover each object’s 3D position and spatial context. More details are provided in Lines 148–159 of the manuscript.
>
> **\[W3\] Can you compare the COCO3D benchmark with other open-vocabulary 3D detection benchmarks?**
>
> Thank you for the insightful suggestion! The table below presents a statistical comparison between COCO3D and existing 3D detection benchmarks on their respective test sets. We report the number of images, instances, categories, covered domains, and the distribution of instances across MS-COCO super categories. As an ongoing effort, we have completed annotation of all COCO validation images since the submission. We refer to this full set as COCO3D v2.
>
> Compared to prior benchmarks—which often focus on specific domains such as indoor environments (e.g., SUN RGB-D), object-centric scenes (e.g., Objectron), or self-driving datasets (e.g., KITTI, nuScenes)—COCO3D offers broader coverage across indoor and outdoor everyday scenes. From the super-category perspective, COCO3D uniquely includes categories from **food**, **animal**, and **sports**, which are often absent in existing 3D datasets. We believe COCO3D can play a similar role in open-vocabulary monocular 3D detection as MS-COCO does in 2D detection—offering a diverse, real-world, and scalable benchmark for advancing the field.
>
> | Dataset | # Images | # Instances | # Categories | Domain | Person | Food | Animal | Vehicle | Kitchen | Furniture | Accessory | Indoor | Sports | Electronic | Outdoor | Appliance |
> |:---:|:---:|:---:|:---:|---|:---:|:---:|:---:|:---:|:---:|:---:|:---:|:---:|:---:|:---:|:---:|:---:|
> | SUN RGB-D | 5,050 | 37,826 | 82 | Indoor | 0.1 | - | - | 0.2 | 6.0 | 65.5 | 8.0 | 11.8 | - | 7.8 | - | 0.5 |
> | ARKitScenes | 7,610 | 56,404 | 15 | Indoor | - | - | - | - | 10.3 | 80.5 | - | 3.1 | - | 4.1 | - | 2.1 |
> | Hypersim | 7,690 | 239,781 | 29 | Indoor | - | - | - | - | 0.9 | 37.2 | 3.6 | 49.7 | - | 8.5 | - | - |
> | Objectron | 9,314 | 10,814 | 9 | Object  | - | - | - | 3.4 | 32.4 | 12.5 | 20.9 | 12.8 | - | 17.9 | - | - |
> | KITTI | 3,769 | 25,909 | 8 | Driving | 12.9 | - | - | 64.4 | - | - | - | - | - | - | 22.8 | - |
> | nuScenes | 6,019 | 57,735 | 9 | Driving | 15.8 | - | - | 63.1 | - | - | - | - | - | - | 21.1 | - |
> | COCO3D v1 | 2,039 | 5,373 | 80 | In the Wild | 27.9 | 9.1 | 10.9 | 11.0 | 8.2 | 4.2 | 4.6 | 4.1 | 4.5 | 4.0 | 1.0 | 2.0 |
> | COCO3D v2 | 4,025 | 18,395 | 80 | In the Wild | 35.6 | 11.6 | 8.8 | 8.2 | 7.4 | 6.6 | 5.2 | 5.0 | 4.7 | 3.4 | 2.3 | 1.3 |
>
>
> **\[Q1\] How to combine MoGe with Depth Pro at Line 141? Why using metric depth estimator there?**
>
> Thank you for the great question\! In our pseudo-annotation pipeline, we align the **relative depth map from MoGe** with the **metric-scale depth map predicted by Depth Pro** using a global affine transformation. Specifically, we first use **MoGe** to generate a relative depth map and extract camera intrinsics. Then, we apply **Depth Pro** to predict a metric-scale depth map for the same scene. We perform RANSAC-based linear regression to fit a robust scale and shift transformation that maps MoGe’s relative depths to Depth Pro’s metric scale. This allows us to retain the fine geometric details from MoGe while calibrating the scale to real-world distances. As a result, our final pseudo 3D annotations are based on **scale-calibrated relative depths**, which we found empirically to improve downstream 3D detection performance compared to using raw MoGe depths alone. This also enables a fairer comparison with baseline methods like **OVM3D-Det**, which rely exclusively on metric depth estimation.
>
> It’s important to note, however, that we do **not** rely on metric depth at evaluation time on the COCO3D benchmark, due to the general difficulty of accurate metric depth prediction in unconstrained scenes. Instead, we introduce a new evaluation metric—**Relative Layout Average Precision** (Lines 235–245)—to assess how well the predicted 3D bounding boxes preserve the relative spatial layout compared to the ground truth.
>
> We will clarify this explanation in the final version of the paper. Thank you again for pointing it out\!
>
> \[1\] Detect Anything 3D in the Wild. Zhang et al. ICCV 2025\.

---

> > ### Author Response · Authors · 2025-08-02
> > **Additional results for DetAny3D (model released after NeurIPS deadline)**
> >
> > Dear reviewer, we present the evaluation results of ICCV 2025 paper DetAny3D in the following table:
> >
> > | | $\text{AP}_{3\text{D}}$ | $\text{AP}_{3\text{D}}^{\text{Rel}}$ |
> > |---|---|---|
> > | Baseline: Omni3D | 5.87 | 20.86 |
> > | DetAny3D | 4.35 | 19.29 |
> > | Omni3D + LabelAny3D | 10.92 | 32.02 |
> >
> > We note that the DetAny3D model checkpoint was only released after the NeurIPS submission deadline (on June 25). To provide a more comprehensive comparison for the reviewers, we took the initiative to implement the evaluation scripts (which the authors had not released) and conducted additional experiments on our COCO3D benchmark with their released model weights.
> >
> > The results indicate that even models that excel in indoor and self-driving domains struggle with in-the-wild scenarios and diverse object categories. This performance gap underscores the critical issue of 3D data scarcity in open-domains.
> > This also highlights the effectiveness of our automated labeling tool and the usefulness of the proposed COCO3D benchmark.
> >
> > Thank you for your constructive review! We hope our detailed explanations have addressed your concerns effectively. Please let us know if you need any further clarification at any point.

---

> > ### Comment · Reviewer_FwMH · 2025-08-06
> >
> > The rebuttal is good. I am willing to accept this paper.

---

> > > ### Author Response · Authors · 2025-08-08
> > >
> > > Thank you for your support and insightful feedback. We will ensure these revisions are included in the final version.

---

> ### Author Response · Authors · 2025-08-05
>
> Dear reviewer,
>
> Thank you again for your thoughtful review and constructive feedback! Your input is invaluable in helping us improve this work. We remain committed to resolving any remaining issues. Please feel free to share any additional comments at your convenience.

---

### Official Review · Reviewer_MxxK · 2025-06-23

**Clarity:** 3
**Significance:** 4
**Originality:** 3
**Rating:** 4
**Confidence:** 4

**Summary:**

This paper proposes an auto labeling framework, called LabelAny3D, for generating 3D bounding box pseudo-labels for in-the-wild 2D images from the MS-COCO dataset. The produced labels are contributed as a new benchmark COCO3D for training and testing open-vocabulary monocular 3D detection systems.

To validate the proposed auto labeling method, the authors compare their approach with an earlier work, OVM3D-Det, in terms of bounding box quality against human labels. In addition, the authors compare the performance of a popular detector, when trained using either the proposed LabelAny3D or OVM3D-Det. Results show that the proposed LabelAny3D yields higher quality bounding boxes than its counterpart over the newly created COCO3D benchmark.

**Questions:**

Please my comments in the weakness section.

**Ethical Concerns:**

["NO or VERY MINOR ethics concerns only"]

**Final Justification:**

After reading additional results and comments from other reviewers, I raise my rating to borderline accept, as my major concern about generalization has been addressed.

**Limitations:**

Yes.

**Quality:**

2

**Strengths And Weaknesses:**

Pros
1. The paper is well written and touches on an important topic with real-world impacts on camera-only autonomous driving and robotics.
2. The proposed auto labeling framework leads to improved bounding box quality compared to a prior art.


Cons
1. The proposed auto labeling pipeline is intuitive and has many hand-engineered modules as well as hyper-parameters. This leads to my concern about the generalization of the proposed approach and sensitivity of those hyper-parameters over a new dataset.
2. Table 1 shows that adding LabelAny3D pseudo labels only leads to marginal performance improvement or degradation over the OVMNono3D Novel and OVMNono3D Base datasets, when comparing against the inclusion of pseudo labels produced by OVM3D-Det.
3. Table 2 only evaluates the two auto labeling methods over the newly proposed COCO3D benchmark, which is susceptible to dataset bias and therefore insufficient. If the quality is indeed improved drastically as listed in the table compared to OVM3D-Det, why does LabelAny3D show mixed results in Table 1? I would suggest the authors run evaluation and compare the two methods over the same datasets employed in the OVM3D-Det paper.


Minor
1. The paper seems incomplete without a conclusion section. Please consider adding that section if accepted.

---

> ### Author Rebuttal · Authors · 2025-07-31
>
> We thank the reviewer for the constructive feedback and for finding our "topic important” and "well written”.
>
> **\[W1\] Concern about the generalization of the proposed approach and sensitivity of the hyper-parameters over a new dataset.**
>
> We thank the reviewer for raising this important concern about generalization and hyperparameter sensitivity. To address this, we have evaluated our pipeline across multiple datasets **without any hyperparameter adjustment**, demonstrating its robustness.
>
> In the original submission, we have validated our pipeline on the **KITTI** benchmark (Lines 280–283). LabelAny3D achieved **13.60 AP3D**, outperforming the baseline OVM3D-Det (**12.39 AP3D**).  In this rebuttal, we provide additional validation on the **SUN RGB-D** benchmark. LabelAny3D surpasses OVM3D-Det by **\+2.32 AP3D** (41.39 vs. 39.07). We will include these results in the final version of the manuscript.
>
> **\[W2\] Adding LabelAny3D pseudo labels only leads to marginal performance improvement or degradation over the OVMNono3D Novel and Base datasets, when comparing against OVM3D-Det.**
>
> Thank you for raising this point\! Both our model and OVM3D-Det are trained on a mix of clean Omni3D labels and generated pseudo labels on COCO, which explains their similar performance on the OVMono3D Novel and Base categories—both derived from Omni3D.
>
> However, our model demonstrates superior performance on COCO3D,  demonstrating the strength of our high-quality pseudo labels in more diverse and open-world settings. Due to the underlying distribution differences, the benefit from our high-quality COCO pseudo annotations to Omni3D is naturally limited. Despite this, we still observe a **\+0.93 AP3D improvement** on OVMono3D Novel categories, while OVM3D-Det shows a **\-0.5 AP3D drop** on OVMono3D Novel categories.
>
> OVM3D-Det shows slightly higher AR3D, which we attribute to overconfident predictions leading to more false positives. We note that average recall (AR3D) captures detection rate but not the accuracy of 3D localization, which is better reflected in AP3D.
>
> We will clarify this point in the revised version.
>
> **\[W3\] Why does LabelAny3D show mixed results in Table 1? … run evaluation and compare the two methods over the same datasets employed in the OVM3D-Det paper.**
>
> Thank you for the constructive suggestion. As noted in our response to W2, while LabelAny3D consistently achieves higher AP3D (precision), OVM3D-Det shows slightly higher AR3D (recall) due to overconfident predictions, which increase false positives. In response to W1, we have also evaluated both methods on the KITTI and SUNRGBD datasets used in the OVM3D-Det paper. These results will be included in the revised version of the manuscript.
>
> **\[Minor\] The paper seems incomplete without a conclusion section. Please consider adding that section if accepted.**
>
> Thank you for pointing this out. Our **Discussion** section (Sec. 5, Lines 304-316) is intended to serve as the conclusion by summarizing our main contributions and acknowledged limitations. For clarity, we will rename it to **Conclusion** in the revised version.

---

> ### Author Response · Authors · 2025-08-05
>
> Dear reviewer,
>
> We appreciate your thoughtful review and constructive suggestions. As the discussion deadline approaches, we would greatly appreciate it if you could let us know whether our responses have fully addressed your concerns. Your feedback is invaluable, and we remain committed to resolving any remaining issues.
>
> To further address the concerns in [W1] and [W3], we additionally validated our approach on nuScenes and Objectron benchmarks. LabelAny3D consistently surpasses OVM3D-Det by +3.16 AP3D on nuScenes (19.50 vs. 16.34) and +2.8 AP3D on Objectron (22.51 vs. 19.71).
>
> We attribute this strong generalizability to two key factors: (1) LabelAny3D effectively integrates multiple 3D foundation models (i.e. TRELLIS, MASt3R, MoGe) which are trained from diverse high-quality 3D data and demonstrate strong generalizability in the wild. This enhances its robustness across datasets and hyperparameters; and (2) LabelAny3D adopts an analysis-by-synthesis paradigm that reconstructs holistic 3D scenes from monocular images in a class-agnostic manner. This enables robust 3D automatic labeling across object categories. In contrast, OVM3D-Det relies on object size priors from large language models, which struggles with categories exhibiting high intra-class variations.
>
> Please don’t hesitate to let us know if further clarification is needed.

---

> ### Author Response · Authors · 2025-08-08
>
> Dear reviewer MxxK,
>
> Thank you for your constructive suggestions! We notice you've submitted your final rating - could you kindly let us know whether our rebuttal addressed your concerns?
>
> In response to your initial comments, we conducted additional experiments on four established 3D benchmarks (including KITTI, SUN RGB-D, nuScenes and Objectron) to address your concerns regarding pipeline robustness and baseline comparisons and observed consistent outperformance. We also provided comprehensive clarifications on the points you raised.
>
> We understand you may have submitted final justifications that are not visible to us during the discussion phase. As this phase comes to a close, we would appreciate any feedback on whether our rebuttal has adequately addressed your concerns. Your feedback is invaluable, and we remain committed to resolving any remaining issues.
>
> We look forward to your reply at your convenience. Thank you again for your time and valuable feedback throughout this process.

---

### Official Review · Reviewer_oo5r · 2025-07-04

**Clarity:** 2
**Significance:** 3
**Originality:** 3
**Rating:** 4
**Confidence:** 3

**Summary:**

This paper proposes LabelAny3D, a framework for efficiently generating high-quality 3D bounding box annotations from 2D images. Based on the LabelAny3D pipeline, the authors introduce COCO3D, a new benchmark for open-vocabulary monocular 3D detection, derived from the MS-COCO dataset. The COCO3D benchmark comprises 2,039 human-refined images with 5,773 instances spanning all 80 categories of MS-COCO. The authors conduct experiments by training monocular 3D detection models on a dataset annotated via the LabelAny3D pipeline, demonstrating that the performance of monocular 3D detectors can be significantly improved.

**Questions:**

1. The author should provide detailed comparision between the proposed dataset and existing open-vocabulary 3D detection.
2. The author should conduct experiments with lareg model architectures to validate the explanations of performance degradation in OVMono3D Novel/base.
3. The author should provide variance analysis across multiple runs on proposed COCO3D benchmark.

**Ethical Concerns:**

["NO or VERY MINOR ethics concerns only"]

**Final Justification:**

This paper proposes a pipeline for generate high-quality 3D bounding box annotations from 2D images and the authors conduct experiments to validate the effectiveness of the proposed method. The rebuttal addressed my concerns, I keep my rating as positive.

**Limitations:**

Yes

**Paper Formatting Concerns:**

No formatting concerns.

**Quality:**

3

**Strengths And Weaknesses:**

Strengths
1. This paper proposes a data processing pipeline to generate pseudo-3D annotations from monocular images. The pipeline leverages several state-of-the-art models (e.g., Depth Pro, Mast3R) to produce plausible 3D bounding boxes.
2. The authors construct the COCO3D dataset, comprising 2,039 human-refined images with 5,373 total instances, establishing a new benchmark for monocular open-vocabulary 3D detection.
3. Experimental results demonstrate that:
(1) The proposed LabelAny3D pipeline achieves remarkable performance improvements compared to state-of-the-art detectors;
(2) Fine-tuning on pseudo-labels generated by LabelAny3D enhances detector performance on the COCO3D benchmark.

Weaknesses
1. Missing detailed comparision between the proposed COCO3D dataset and other exist benchmark dataset for open-vocabulary 3D detection. While the authors claim existing benchmarks lack diversity for open-world scenarios, they fail to provide a detailed quantitative comparison between COCO3D and established datasets. A systematic comparison should include: (1) Dataset scale (number of images/instances); (2) Category coverage and diversity;  and (3) Scenario variety.
2. Finetuning with the pseudo-label generated by LabelAny3D achieve not significant performance improvement even with degration in OVMono3D Novel and OVmono3D Base, the author explain that the eason can be attribute to (1) limited model capacity and (2) label noise. The author should provide a experiments with larger model capacity to validate. Without such evidence, the explanations remain speculative.
3. The proposed COCO3D only consist of 5373 instances. The statistical significance for reliable evaluation is questionable.

---

> ### Author Rebuttal · Authors · 2025-07-31
>
> We appreciate the reviewer for the thoughtful feedback and praises, including "remarkable performance improvements" and "enhances detector performance".
>
> **\[W1 & Q1\] Systematic comparison of COCO3D benchmark with other benchmarks.**
>
> We thank the reviewer for suggesting this important point\! The table below presents a statistical comparison between COCO3D and existing 3D detection benchmarks on their respective test sets. We report the number of images, instances, categories, covered domains, and the distribution of instances across MS-COCO super categories. As an ongoing effort, we have completed annotation of all COCO validation images since the submission. We refer to this full set as COCO3D v2.
>
> Compared to prior benchmarks—which often focus on specific domains such as indoor environments (e.g., SUN RGB-D), object-centric scenes (e.g., Objectron), or self-driving datasets (e.g., KITTI, nuScenes)—COCO3D offers broader coverage across indoor and outdoor everyday scenes. From the super-category perspective, COCO3D uniquely includes categories from **food**, **animal**, and **sports**, which are often absent in existing 3D datasets. We believe COCO3D can play a similar role in open-vocabulary monocular 3D detection as MS-COCO does in 2D detection—offering a diverse, real-world, and scalable benchmark for advancing the field.
>
> | Dataset | # Images | # Instances | # Categories | Domain | Person | Food | Animal | Vehicle | Kitchen | Furniture | Accessory | Indoor | Sports | Electronic | Outdoor | Appliance |
> |:---:|:---:|:---:|:---:|---|:---:|:---:|:---:|:---:|:---:|:---:|:---:|:---:|:---:|:---:|:---:|:---:|
> | SUN RGB-D | 5,050 | 37,826 | 82 | Indoor | 0.1 | - | - | 0.2 | 6 | 65.5 | 8.0 | 11.8 | - | 7.8 | - | 0.5 |
> | ARKitScenes | 7,610 | 56,404 | 15 | Indoor | - | - | - | - | 10.3 | 80.5 | - | 3.1 | - | 4.1 | - | 2.1 |
> | Hypersim | 7,690 | 239,781 | 29 | Indoor | - | - | - | - | 0.9 | 37.2 | 3.6 | 49.7 | - | 8.5 | - | - |
> | Objectron | 9,314 | 10,814 | 9 | Object  | - | - | - | 3.4 | 32.4 | 12.5 | 20.9 | 12.8 | - | 17.9 | - | - |
> | KITTI | 3,769 | 25,909 | 8 | Driving | 12.9 | - | - | 64.4 | - | - | - | - | - | - | 22.8 | - |
> | nuScenes | 6,019 | 57,735 | 9 | Driving | 15.8 | - | - | 63.1 | - | - | - | - | - | - | 21.1 | - |
> | COCO3D v1 | 2,039 | 5,373 | 80 | In the Wild | 27.9 | 9.1 | 10.9 | 11.0 | 8.2 | 4.2 | 4.6 | 4.1 | 4.5 | 4.0 | 1.0 | 2.0 |
> | COCO3D v2 | 4,025 | 18,395 | 80 | In the Wild | 35.6 | 11.6 | 8.8 | 8.2 | 7.4 | 6.6 | 5.2 | 5 | 4.7 | 3.4 | 2.3 | 1.3 |
>
> **\[W2 & Q2\] Analysis of larger model performance on COCO3D benchmark**
>
> We thank the reviewer for suggesting experiments with larger model capacity. The results are provided in the following table. Both the baseline OVMono3D and our model were initially trained with a frozen DINOv2 backbone. To increase model capacity, we unfroze the DINOv2 backbone for both models, thereby introducing more learnable parameters.
>
> After enlarging the model capacity, both the baseline and our model demonstrated performance improvements. This indicates that the backbone produces more task-related features after unfreezing. More importantly, the performance gap between our model and OVMono3D on base categories narrowed from 2 to 1.3 AP points. This suggests that with larger model capacity, models can better handle more diverse scenarios and categories, thus alleviating the catastrophic forgetting issue.
>
> We also attempted to enlarge model capacity by replacing the backbone from DINOv2-base with DINOv2-large. However, the required training time exceeds the rebuttal period given our computational resources. We will strive to include this analysis in our revised version.
>
> | Method | Model Capacity | $\text{AP}_{3\text{D}}$ COCO3D | $\text{AP}_{3\text{D}}^{\text{Rel}}$ COCO3D | $\text{AP}_{3\text{D}}$ OVMono3D Novel | $\text{AP}_{3\text{D}}$ OVMono3D Base |
> |:---:|:---:|:---:|:---:|:---:|:---:|
> | Baseline: Omni3D | Standard | 5.87 | 20.86 | 16.05 | 24.77 |
> | Baseline: Omni3D | Large | 6.47 | 22.36 | 17.05 | 24.87|
> | Omni3D + LabelAny3D | Standard | 10.92 | 32.02 | 16.98 | 22.74 |
> | Omni3D + LabelAny3D | Large | 11.94 | 33.44 | 17.08 | 23.54 |
>
>
> **\[W3 & Q3\] Variance analysis across multiple runs on proposed COCO3D benchmark.**
>
> We appreciate the reviewer's constructive suggestion\! We conducted three training runs for each model in Table 1 (except for the baseline, for which we directly evaluated its publicly-released model checkpoint). The evaluation results on COCO3D are shown in the following table, with data presented in mean ± standard deviation format. These results validate that our benchmark provides consistent and reliable model rankings despite the inherent randomness in training procedures.
>
>
> |  | $\text{AP}_{3\text{D}}$ | $\text{AP}_{3\text{D}}^{\text{Rel}}$ |
> |---|---|---|
> | Baseline: Omni3D |5.87  | 20.86 |
> | OVM3D-Det | 2.68 ± 0.14 |8.17 ± 0.14  |
> | Omni3D + OVM3D-Det | 6.68 ± 0.27 | 19.92 ± 0.55 |
> | LabelAny3D | 7.68 ± 0.13 | 24.64 ± 0.08 |
> | Omni3D + LabelAny3D | 10.92 ± 0.36 | 32.12 ± 0.26 |
>
>
> We also conducted multiple evaluation runs for each model using different subsets of COCO3D. Specifically, we sampled three subsets from the COCO3D benchmark, each containing 1500 images. Each model in Table 1 was evaluated on each subset separately, and we report the mean ± standard deviation of AP points in the following table. The results across the three subsets are stable, which further demonstrates the robustness of our benchmark.
>
> |  | $\text{AP}_{3\text{D}}$ | $\text{AP}_{3\text{D}}^{\text{Rel}}$ |
> |---|---|---|
> | Baseline: Omni3D |6.10 ± 0.18 | 21.51 ± 0.73 |
> | OVM3D-Det |2.81 ± 0.13  |8.20 ± 0.33  |
> | Omni3D + OVM3D-Det |7.03 ± 0.40  | 21.31 ± 0.42 |
> | LabelAny3D |8.21± 0.46  | 25.32 ± 0.37 |
> | Omni3D + LabelAny3D |11.28 ± 0.62 |32.59 ± 1.46 |
>
>
> We agree that including more images and instances would enhance the reliability and robustness of the benchmark. Since the time of submission, we have completed annotations for the entire MS-COCO validation set, and we plan to incorporate additional datasets such as Objects365 to further expand the benchmark.

---

> ### Author Response · Authors · 2025-08-05
>
> We sincerely appreciate your valuable feedback and support. These revisions will be fully incorporated into our final version.

---

### Official Review · Reviewer_huQ4 · 2025-07-04

**Clarity:** 3
**Significance:** 3
**Originality:** 3
**Rating:** 5
**Confidence:** 4

**Summary:**

This paper proposes an annotation pipeline to annotate the 3D bounding boxes given 2D images. Specifically, the authors propose an analysis-by-synthesis framework that first reconstructs holistic 3D scenes from 2D images and then recovers the 3D bounding boxes with absolute scale. Along with the proposed pipeline and benchmark, the authors also train a monocular 3D object detector based on the dataset, which shows superior performance on the designed COCO3D benchmark.

**Questions:**

Regarding object orientation, how should the proposed pipeline handle round objects that may lack a clear definition of orientation?

**Ethical Concerns:**

["NO or VERY MINOR ethics concerns only"]

**Final Justification:**

The author's response addresses most of my concerns. I will keep my positive rating.

**Limitations:**

yes

**Quality:**

4

**Strengths And Weaknesses:**

### Strength
1. The writing is clear, and the paper is easy to follow.
2. The proposed annotation pipeline is theoretically significant, as stated in the paper, it enables labeling of any 3D objects from images.
3. According to the visualizations in the paper, the annotation quality is excellent, and the model trained on the datasets demonstrates strong 3D generalization capability on the COCO dataset.


### Weakness
1. The overall annotation pipeline lacks rigorous evaluation. The authors could validate the annotated 3D bounding boxes on datasets with 3D ground truth from LiDAR data (e.g., KITTI, nuScenes, or ScanNet).

---

> ### Author Rebuttal · Authors · 2025-07-31
>
> We thank the reviewer for the constructive comments and for recognizing our “clear writing”, “theoretically significant”, and “strong 3D generalization capability".
>
> **\[W1\] Validation of the pipeline on datasets with 3D ground truth.**
>
> We thank the reviewer for raising this important point. In the original submission (Lines 280–283), we validated our pipeline on the **KITTI** benchmark, where LabelAny3D achieved **13.60 AP3D**, outperforming the baseline OVM3D-Det (**12.39 AP3D**). In this rebuttal, we provide additional validation on the **SUN RGB-D** benchmark. LabelAny3D surpasses OVM3D-Det by **\+2.32 AP3D** (41.39 vs. 39.07). OVM3D-Det relies on object size priors from large language models to infer 3D bounding boxes. This strategy is particularly effective for objects with relatively consistent dimensions (e.g., cars), but struggles with categories exhibiting high intra-class variation (Lines 41-45). In contrast, LabelAny3D adopts an analysis-by-synthesis paradigm: it reconstructs the holistic 3D scene from monocular images and uses the synthesized representation to infer 3D annotations (Lines 52-54). We will include these results and extend our validation to more benchmarks in the final version of the manuscript.
>
> **\[Q1\] How should the proposed pipeline handle round objects that may lack a clear definition of orientation?**
>
> We thank the reviewer for this insightful question\! Orientation ambiguity in symmetric objects is a well-known challenge in 3D vision. For such objects, multiple orientations may yield indistinguishable appearances, making pose estimation inherently ambiguous.
> Our pipeline does not explicitly resolve this ambiguity. Instead, it inherits the pose definition from TRELLIS, the 3D reconstruction model we used, which reconstructs objects in the canonical space defined in its training data, where the orientation of symmetric objects remains ill-defined.
>
> Some recent works (e.g., Orient Anything \[1\]) address this issue by using vision-language models (VLMs) to identify symmetric objects and label them as having no meaningful pose. However, our primary focus is on tight 3D bounding box prediction, which remains meaningful even for symmetric objects. We therefore retain symmetric objects in both training and evaluation to preserve dataset diversity. This approach aligns with standard practice—most large-scale 3D datasets (e.g., KITTI, ARKitScenes) do not explicitly handle symmetry during training or evaluation.
>
> An exception is Objectron \[2\], which incorporates pose equivalence in its evaluation metrics to account for such ambiguity. Specifically, predicted bounding boxes for symmetric objects are rotated along the symmetry axis, and the rotation that yields the maximum IoU with the ground truth is selected.
>
> To assess the impact of symmetry on our benchmark, we manually annotated symmetric instances in 300 randomly sampled images  from COCO3D. We found that symmetric objects comprise 13% of all instances, primarily in cylindrical shapes such as apples, bottles, and bowls.  Using the pose-equivalent metric from Objectron, our 3D detector achieves **12.55 AP3D** on symmetric objects, compared to **12.53 AP3D** without the metric—a negligible difference of **0.02 AP3D**.
>
> In summary, our results suggest that our current results are robust to orientation ambiguity. We will include a detailed discussion of this limitation and our evaluation results in the final version of the paper.
>
> \[1\] Orient Anything: Learning Robust Object Orientation Estimation from Rendering 3D Models. Wang et al. ICML 2025\.
> \[2\] Objectron: A Large Scale Dataset of Object-Centric Videos in the Wild with Pose Annotations. Ahmadyan et al. CVPR 2021\.

---

> ### Author Response · Authors · 2025-08-05
>
> Dear reviewer,
>
> Your constructive feedback has strengthened our evaluation, and we sincerely appreciate it!
>
> To further address the concerns in [W1], we additionally validated our approach on nuScenes and Objectron benchmarks. Consistent with our prior results, LabelAny3D surpasses OVM3D-Det by +3.16 AP3D on nuScenes (19.50 vs. 16.34) and +2.8 AP3D on Objectron (22.51 vs. 19.71). By reconstructing the entire scene, LabelAny3D effectively works across different benchmarks without relying on per-category priors.
>
> We hope our responses have adequately addressed your concerns, but please don't hesitate to point out any remaining issues. Your input is invaluable in helping us improve this work.

---

> > ### Comment · Reviewer_huQ4 · 2025-08-06
> > **Reply by reviewer**
> >
> > The response addresses most of my concerns. I will keep my positive rating.

---

> > > ### Author Response · Authors · 2025-08-08
> > >
> > > Thank you for your support and insightful feedback. We will ensure these revisions are included in the final version.

---

### Note · Authors · 2025-08-11

We thank all reviewers for their constructive feedback. The reviewers recognized our work's theoretical significance, strong 3D generalization capability, clear writing, and novel pipeline design. We appreciate the unanimous positive responses.

**Main Contributions:**

Our work tackles the fundamental challenge of 3D object detection in open-world scenarios. While existing models perform well in indoor/driving domains, they fail with in-the-wild images due to limited 3D datasets. We contribute: **(1) LabelAny3D**: an efficient analysis-by-synthesis framework that reconstructs holistic 3D scenes to generate high-quality 3D bounding box annotations; **(2) Performance validation** showing our annotations consistently improve monocular 3D detection, outperforming existing auto-labeling approaches; **(3) COCO3D benchmark**: featuring diverse object categories for open-vocabulary monocular 3D detection.

**Key Contributions Validated Through Rebuttal:**

**1. Robust Generalization:** Extensive validation across KITTI (+1.21 AP3D), SUN RGB-D (+2.32 AP3D), nuScenes (+3.16 AP3D), and Objectron (+2.8 AP3D) shows consistent LabelAny3D improvements over baseline without hyperparameter tuning.

**2. Comprehensive Benchmark:** COCO3D offers unprecedented diversity with 80 categories across indoor/outdoor scenes, uniquely covering food, animal, and sports categories missing from existing 3D datasets.

**3. Analysis-by-Synthesis Approach:** Our holistic 3D scene reconstruction fundamentally differs from size-prior methods, achieving robust performance across object categories with high intra-class variation.

**Concerns Addressed Through Rebuttal:**

- **Orientation Ambiguity:** Evaluation on symmetric objects (13% instances) shows negligible impact (0.02 AP3D difference)
- **Model Capacity:** Larger models reduce performance gaps, validating scalability
- **Variance Analysis:** Multiple runs verify consistent model rankings and benchmark reliability
- **Additional Baselines:** DetAny3D (ICCV 2025) evaluation (4.35 vs. our 10.92 AP3D) further confirms our approach's effectiveness

Reviewers provided positive feedback: huQ4 and oo5r stated "will keep my positive rating" with "concerns addressed," while FwMH declared "willing to accept." We hope our detailed response has also addressed concerns from Reviewer MxxK. We commit to incorporating all suggested improvements in the final version.

Thank you again for your time and effort. We look forward to your final decision.

---

### Decision · Program_Chairs · 2025-09-17

**Decision:**

Accept (poster)

**Comment:**

Based on a thorough review of the submission, the reviewer discussions, and the authors' rebuttal, AC recommends accepting this paper, mainly due to the contribution of the new open-vocabulary monocular 3D detection benchmark. The work introduces LabelAny3D, a novel and effective analysis-by-synthesis framework for generating high-quality 3D bounding box annotations from 2D images, and presents the COCO3D benchmark to advance open-vocabulary 3D detection. Reviewers unanimously praised the paper's clear writing, the significance of tackling 3D detection in the wild, and the strong potential impact of the proposed annotation pipeline and dataset. Initial concerns regarding the pipeline's generalization, the lack of comparison to existing benchmarks, and the breadth of experimental validation were comprehensively addressed by the authors during the rebuttal period. The authors provided extensive new experiments across four additional established benchmarks, demonstrating consistent performance gains without hyperparameter tuning and thus validating the method's robustness. They also provided detailed quantitative comparisons of their new benchmark and addressed questions about model capacity and evaluation on symmetric objects. The thorough rebuttal successfully resolved the reviewers' concerns, leading all of them to converge on a positive recommendation for acceptance.